# Machine learning algorithm for ventilator mode selection, pressure and volume control

**Anitha T.** [1]*, **Gopu G.**[2], **Arun Mozhi Devan P.**[3], **Maher Assaad**[4]*

**1** Department of Electronics and Instrumentation Engineering, Sri Ramakrishna Engineering College, Coimbatore, Tamil Nadu, India, **2** Department of Electronics and Communication Engineering, Sri Ramakrishna Engineering College, Coimbatore, Tamil Nadu, India, **3** Department of Electrical and Electronics Engineering, Universiti Teknologi PETRONAS, Seri Iskandar, Malaysia, **4** Department of Electrical and Electronics Engineering, Department of Electrical and Computer Engineering, Ajman University, Ajman, United Arab Emirates

* anithacie@srec.ac.in (AT); m.assaad@ajman.ac.ae (MA)

## Abstract

Mechanical ventilation techniques are vital for preserving individuals with a serious condition lives in the prolonged hospitalization unit. Nevertheless, an imbalance amid the hospitalized people demands and the respiratory structure could cause to inconsistencies in the patient's inhalation. To tackle this problem, this study presents an Iterative Learning PID Controller (ILC-PID), a unique current cycle feedback type controller that helps in gaining the correct pressure and volume. The paper also offers a clear and complete examination of the primarily efficient neural approach for generating optimal inhalation strategies. Moreover, machine learning-based classifiers are used to evaluate the precision and performance of the ILC-PID controller. These classifiers able to forecast and choose the perfect type for various inhalation modes, eliminating the likelihood that patients will require mechanical ventilation. In pressure control, the suggested accurate neural categorization exhibited an average accuracy rate of 88.2% in continuous positive airway pressure (CPAP) mode and 91.7% in proportional assist ventilation (PAV) mode while comparing with the other classifiers like ensemble classifier has reduced accuracy rate of 69.5% in CPAP mode and also 71.7% in PAV mode. An average accuracy of 78.9% rate in other classifiers compared to neutral network in CPAP. The neural model had an typical range of 81.6% in CPAP mode and 84.59% in PAV mode for 20 cm $H_2O$ of volume created by the neural network classifier in the volume investigation. Compared to the other classifiers, an average of 72.17% was in CPAP mode, and 77.83% was in PAV mode in volume control. Different approaches, such as decision trees, optimizable Bayes trees, naive Bayes trees, nearest neighbour trees, and an ensemble of trees, were also evaluated regarding the accuracy by confusion matrix concept, training duration, specificity, sensitivity, and F1 score.

## 1 Introduction

To understand the how an automatic ventilator performs while artificial airflow is being provided, a mathematical model may be created and utilized to simulate various circumstances [1,

---

**Data Availability Statement:** All relevant data are within the manuscript and its Supporting information files.

**Funding:** The authors received no specific funding for this work.

**Competing interests:** The authors have declared that no competing interests exist.

**Abbreviations:** CMV, Continuous Mandatory Ventilation; CPAP, Continuous Positive Airway Pressure; DT, Decision Tree; ET, Ensemble Classifier; FN, False Negative; FP, False Positive; ILC, Iterative Learning Controller; ML, Machine Learning; NB, Naive Bayes; NBT, Naive Bayes Tree; NeNT, Nearest neighbour Tree; NN, Neural Network; NNC, Neural Network Classifier; NNT, Neural Network Tree; OBT, Naive Bayes of Optimizable Bayes Tree; PAV, Proportional Assist Ventilation; ROC, Region of Convergence; SIMV, Synchronized Intermittent Mandatory Ventilation; TN, True Negative; TP, True Positive.

2]. A pressure-controller ventilator was used to develop and simulate a mathematical Signal driven convolutional auto-encoder model. The model was created to assess the degree of asynchronous Inhaling while using supplemental airflow, and it endured training on 400,000 instances to recognize regular breathing routines. ABReCA increases comprehension of patient-ventilator interaction quality by providing more accurate assessments [3]. By changing the flow given by the ventilator, modular PID controller keeps maximum pressure lower than necessary [4]. The airflow control is done with improved precision on a first-order non-linear structure based on interruptions in time utilizing fractional-order controllers [5]. Researchers in [6], a machine learning technique has been developed by investigators that employs waveform analysis to accurately identify instances of patient ventilator inconsistency. Furthermore, the framework identifies cycle asynchrony and possesses a high degree of sensitivity and specificity in determining its presence, rendering it a trusted tool.

Zhang et al. created a method for precise blood flow calculation using photoplethysmography (PPG), a valid clinical approach [7]. The authors' work demonstrates the efficacy of machine learning algorithms in tackling the issues involved with broad arterial pressure monitoring. The results of this study are promising, as machine learning techniques can provide high accuracy in blood pressure estimation. Moreover, Zhang et al. conducted research demonstrating the efficacy of neural network-based machine-learning models in identifying patient-ventilator asynchrony during mechanical breathing [8]. The suggested neural network model has improved in terms of resilience and consistency. The model's categorization method has also been enhanced to assist physicians in better grasping the outcomes of deep learning technology. Based on the findings, machine learning approaches can considerably increase the interpretability of categorization results, allowing physicians to make better-educated judgments. In [9], The study used machine learning to detect clinical characteristics in critically sick patients on respiration in emergency. A individual automatic lung situation, variables are decided based on inspection and the person's engage; these factors vary based on breathing status. Using appropriate data, the machine learning algorithm can forecast criteria for artificial respiration for numerous kinds of respiratory illnesses.

Investigators have established a compelling neural network model that can classify breaths individually. This enables the medical assistance for inhalation to employ perfect respiratory form predictions [10, 11]. In [12, 13], research papers use real patient information and apply a nearest neighbor algorithm and categorization approaches to artificial intelligence to anticipate variations in respiratory mode in after surgery patients during the weaning phase. The single-compartment linear computational representation approximates a human's resting breathing cycle. Recent research by Zahia et al. In [14], shows that machine learning and intense learning can accurately diagnose bruises from pressure and follow their recovery process. Furthermore, the study underlines the need to segment the wound correctly, identify its material kinds, and monitor its recovery progress.

Richard et al. introduced the xPULM breathing process, which integrates mathematical and mechanical models to recreate the human respiratory system's complicated non-linear patterns properly. The simulator replicates needed breathing patterns with excellent conformance and flexibility, and it also serves as a link between airway models that are biomechanical and mathematical [15]. Mehedi et al. in [16] suggested a control technique that employs alternating law to regulate in accordance with the shifting behaviour concept to reduce the distinction among full output linearization control and an estimated singular fuzzy motion. Despite the absence of leak characteristics and pulmonary comfort of a sick person, this technique helps manage the respiratory airway flow and inhibits maximum pressure from reaching crucial levels. While maintaining the required airway pressure, the modelling outcomes reveal quicker conjunction, reduced small tracking mistakes and overstepping. Arun et al. in [17, 18]

undertake a similar type of case study to manage the airflow and maintain pressure utilizing various optimization strategies and fractional-order control for successful inactive compensation. The suggested method provides enough sorting while executing indeed in set point and disturbance procedures and managing overshoot.

Create concepts for a remote surveillance framework customized to individuals with numerous prolonged diseases were identified using machine learning applications: modularity, regular reinforcement, record consolidation, and beneficial critique. It is challenging to focused on the victim mobile application for numerous prolonged illnesses, however, these guidelines may help the project, and adding patient and physician feedback is critical [19–21]. Deep learning approaches have shown a great capacity to predict and identify PVA. However, these techniques are constrained since model training requires a large quantity of labelled information, which might impede practical application [22, 23]. A framework for transfer learning has been created according to pre-defined neural model tailored particularly for asynchrony support using limited data records. A portion of descent cross-validation approach has been devised to test design functionality on short records. The effects show that their suggested transfer learning technique delivers good PVA detection accuracies when dealing with small datasets [13, 24].

Peine et al. presented a method for optimizing breathing techniques for critically sick patients using reinforcement learning algorithms [25]. The objective is to give repeatable high performance by adopting a personalized approach dynamically. The suggested algorithm beat physicians' usual clinical care on two datasets. The algorithm, in particular, typically selected ventilation parameters with lower tidal volume and PEEP levels without exceeding a significant percentage of infused oxygen consumption. This method can potentially enhance patient outcomes and might be viable alternative to standard clinical care.

A detailed examination machine learning techniques, especially K-Nearest Neighbour, Genetic method, Support Vector model, Decision categorization, and Long Short Term Memory, is performed for practical records and forecasting insights. The research extensively investigates each method preciseness, robustness, and dependability and gives comparative and evaluative conclusions. The investigation demonstrates that the long short-term memory (LSTM) and support vector method outperform the other techniques [26, 27]. Yu et al. [28, 29] created a artificial model for forecasting the mortality rate in the hospital from corona patients on ventilatory support. Their technique employed a couple of distinct methods were used to instruct and verify the cohorts. Simple criteria such as age and functions are used to forecast the requirement for artificial breathing, assisting ER physicians in deciding whether to admit the patient to the hospital or discharge the individuals residence. The findings suggest that machine learning algorithms successfully forecast respiration and death requirements in patients of corona disease.

Table 1 provides an overall overview for several discriminant settings approach with distinct methods and their optimal dimension for ventilating devices. The following are the research article's crucial contributions:

1. An utterly novel concept To attain the desired pressure and volume, an iterative learning PID controller (ILC-PID) is presented.

2. Using several machine learning-based classifiers, anticipate the optimum functioning model for the numerous respirator configurations.

3. Data pre-processing comprises seventy percent training data and thirty percent evaluation data, and the study employs a variety of classifier methods

4. The suggested method is tested for performance aspects such as accuracy, training duration, sensitivity, specificity, F1 score, and precision in 20 cm $H_2O$ by cross-validation strategy.

**Table 1. The findings of a meta-analysis of various mechanical ventilators with the suggested model.**

| Ref. | Model Type | Size Effective (95% The Interval of Confidence) | | | |
|------|-----------|------------|------------|------------|------------|
| | | Accuracy | Sensitivity | Specificity/Precision | F1score |
| [30] | Convolutional LSTM | 0.99 (0.99–1.0) | 0.99(0.94–1.0) | 0.90 (0.84–1.01) | 0.93 (0.935–1.0) |
| [31] | Random forest | 0.89 (0.098–1.01) | 0.90 (0.118–1.01) | 0.90 (0.876–0.924) | 0.90 (0.98–1.01) |
| [6] | Machine learning | 0.94 (0.80–0.98) | 0.97 (0.98–1.01) | 0.93 (0.9 -1.04) | 0.94 (0.8–1.02) |
| [32] | Deep neural network | 0.98 (0.81 – 0.98) | 0.98 (0.82–1.01) | 0.87 (0.71–1.03) | 0.89 (0.85–1.05) |
| **Suggested** | **Neural network** | **0.91 (0.7- 0.89)** | **0.91 (0.71–0.87)** | **0.91 (0.7–1.02)** | **0.95 (0.84–1.1)** |

5. The suggested study was carried out in simulation, and the outcomes were obtained using supervised classifier approaches with ILC-PID Controller controllability.

6. From the data, the utmost perfect neural model had an median preciseness of 88.2% in CPAP mode and 91.7% in PAV mode at 20 cm $H_2O$ of pressure by using confusion matrix calculation.

Most current ventilators have different modes to choose from, all of which rely on intermittent positive-pressure breathing. The most often used modes are continuous mandated ventilation (CMV), which includes both pressure control (PCMV) and volume control (VCMV), and synchronized intermittent mandatory ventilation (SIMV), which includes both pressure and volume control. It is essential to be aware that CMV can result in over-ventilation if the respiratory rate is too high or if the patient's spontaneous respiration attempts are not in harmony with the ventilator. Additionally, setting pressure restrictions too high in the event of PCMV can lead to over-distension of the alveoli, which may cause lung damage. High airway pressures, especially when lung compliance changes, may result from VCMV, which increases the risk of barotrauma (lung damage due to increased pressure). SIMV may need to be more adaptable to rapidly adjust to changes in a patient's respiratory demands than other ventilation modalities. In the event of a modify in the person respiration factors, it may be mandatory to modify the inhalation settings. Avoid the above-mentioned reason because, while it is listed as often utilized, these modes have the constraint of lower airway pressure. CPAP that automatically adjusts the patient's pressure. PAV is defined as ventilatory assistance provided in response to recognized abnormalities in respiratory function. This study concentrated on two modes: continuous positive airway pressure (CPAP) and proportional assist ventilation (PAV). Knowing about these conventional modes will help you comprehend their importance and the variations in operation between each mode. This created the need to apply various artificial intelligence approaches to forecast the optimal performing models that utilize various ventilator settings when there is ILC-PID control. Fig 1 depicts the general goal of this study and use of the suggested neural network classifier (NNC) method for respiratory choice of function.

The proposed work is organized as subsequently: The second portion contains extensive simulation of breathing machine. The third portion describes the data creation and artificial intelligence based classifier algorithms in depth also described about the metrics for classification. The fourth portion describes an iterative learning controller-based PID controller architecture. The fifth portion presents the findings of several classifier methods for respiratory pressure and volume evaluation in different formats. The sixth portion includes a summary and closing remarks. Finally, all abbreviated words used in the article are mentioned in the 6.2.

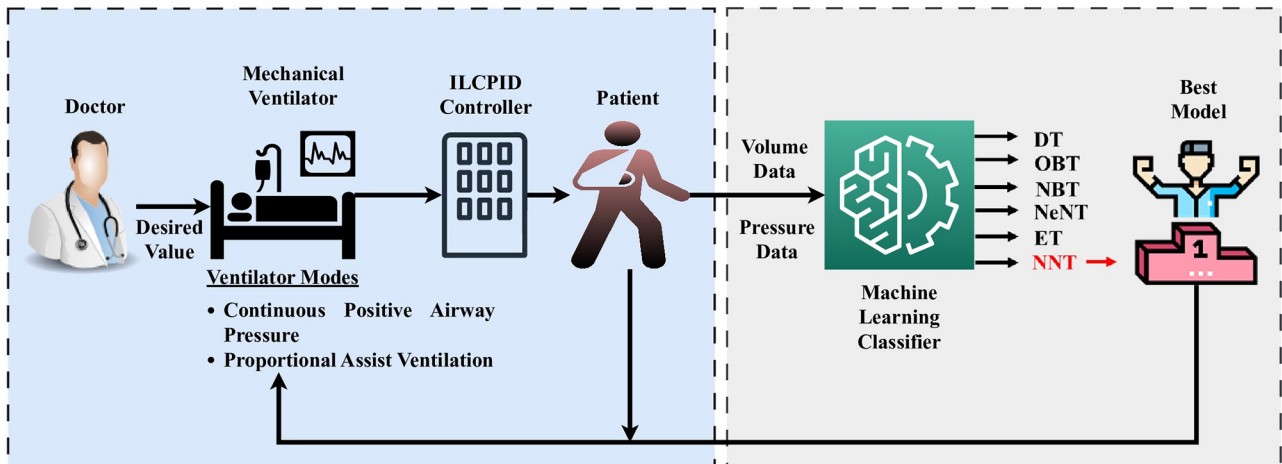

**Fig 1. General objective and research flow of the article.**

## 2 Modelling

Fig 2(a) depicts the ventilator machine modelling. During the initial breath, the prototype depicts the regulated volume and pressure of the respiration. The airflow rate must obey the relation in the volume-controlled condition, according to Eq 1.

$$q_{\text{ref}} = \begin{cases} \gamma, & \text{if } \gamma t < VT \text{ and } 0 < t < t_{\text{insp}} \\ 0, & \text{otherwise} \end{cases} \tag{1}$$

where as $\gamma$ denotes the appropriate airflow throughout inspiratory and $VT$ indicates the tidal volume. The variables utilized in modelling the breathing apparatus are shown in Table 2. The matching movement of electrical networks and pressure presentation as current and voltage from Fig 2(a) is shown in Fig 2(b). These solenoid devices were allocated the function of resistors controlled by resistors disabled by voltage. The valve resistances are $Rv_1 max$ and $Rv_2 max$ while the valves are shut and $Rv_1 min$ and $Rv_2 min$ when the actuators are released. The pulse width modulator controls the solenoid pump, and the mean pressure may be read as a continuously shifting signal with a suitably increased frequency switchback. The ventilator's finalized modified analog circuits shown in Fig 2(c). The patient's comparable simplified circuit inspiration cycle is depicted in Fig 2(b). A similar transfer function is described in the following equation.

$$G(s) = \frac{Q_0(s)}{V(s)} = \frac{C_c}{C_c C_2 R_{2c} s + C_c + C_2} e^{-0.45 T_{\text{resp}} \; s} \tag{2}$$

where $Q_0(s)$ represents the air flow rate. The volume rate is denoted by the symbol $V(s)$. Applying all parameters in the preceding equation results in the final transfer function $G_1(s)$, as shown in Eq 3.

$$G_1(s) = \frac{0.07425}{0.022s + 0.54} \tag{3}$$

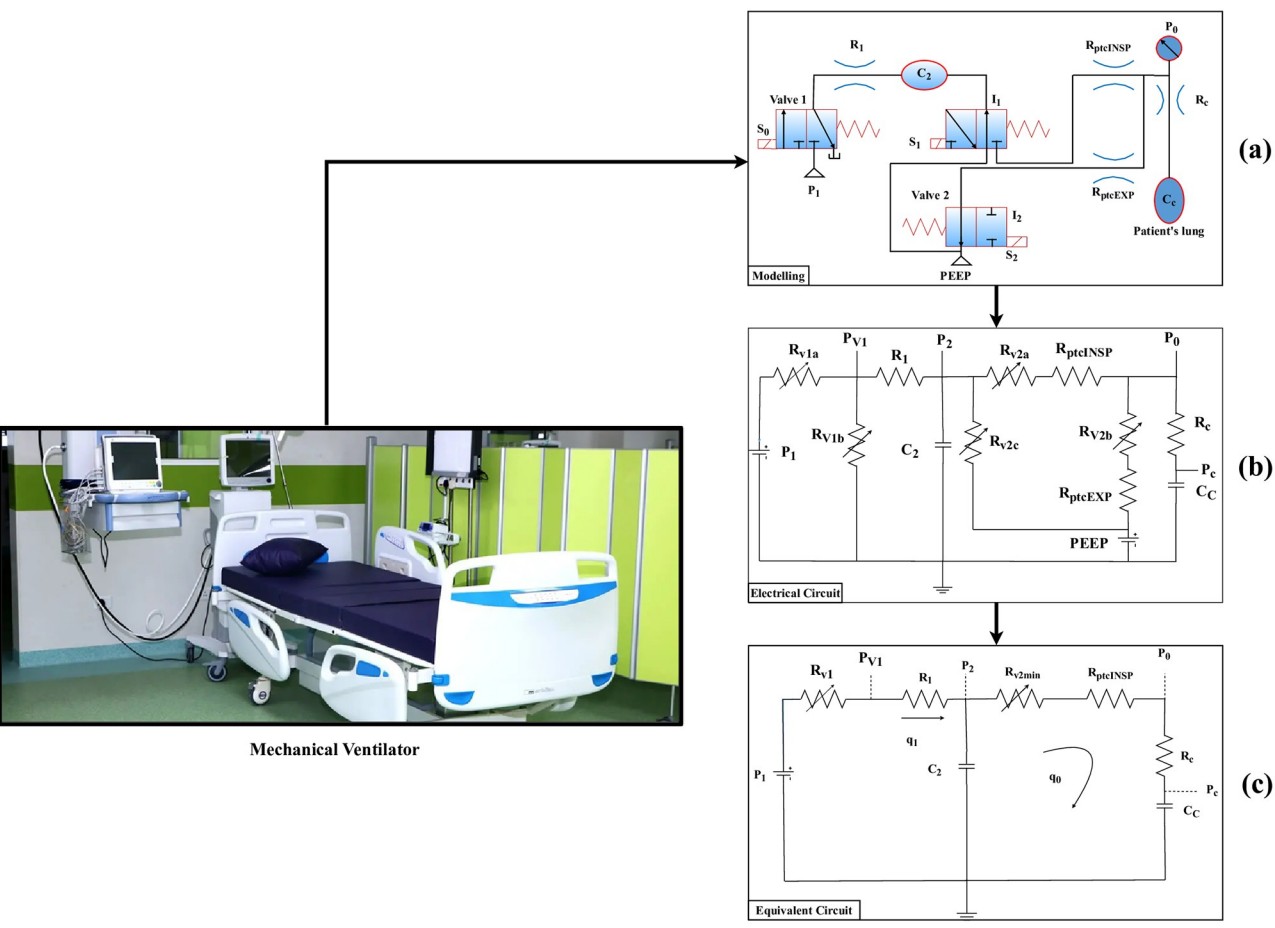

**Fig 2. Modelling of ventilator system.**

**Table 2. Modelling parameters for a ventilator system.**

| Parameter | Definition |
|---|---|
| $C_2$ | Load pressure reservoir (0.465 ml/cm $H_2O$) |
| $C_c$ | Pulmonary compliance (0.075 ml/cm $H_2O$) |
| $I_1$ | 3-way two position valve |
| $I_2$ | 2-way two position valve |
| PEEP | Pressure of water based column |
| $P_1$ and $P_2$ | Pressure sources |
| $P_o$ | Pressure transducer |
| $q_1$ and $q_o$ | Volumetric flow |
| $R_1$ | Pneumatic resistance |
| $R_c$ | Patience airway resistance (0.377 cm $H_2O$/ml/s) |
| $R_{v2min}$ | Voltage controlled resistance (1.062 × $10^{-4}$ cm $H_2O$/ml/s) |
| $R_{v1}$, $R_{v2c}$, $R_{v2a}$ and $R_{v2b}$ | Variable resistance |
| $R_{ptpcINSP}$ and $R_{ptpcEXP}$ | Inspiration and expiration resistance |
| $S_0$, $S_1$ and $S_2$ | Solenoid valves |
| $T_{resp}$ | Breathing cycle of time Period |

## 3 Formation of data

It is critical to acquire appropriate data to develop a successful machine-learning model. This research looked at data from breathing machines and patient medical records stored online. Non-invasive measurements derived from regular medical records and ventilators included 28 aspects. Simultaneously data were obtained from manually by medical apparatus, medicine, and nursing. Age, gender, discharge of urine, and laboratory values were among the remaining factors. A multi-class classification strategy correctly captured patient-specific features throughout the data production phase. The suggested neural network classifier, on the other hand, employs binary categorization, with volume allocated to class 0 and pressure attributed to class 1. More research is needed to develop alternate techniques for adding new individual characteristics into the categorization process. The information for this study was acquired from Sri Ramakrishna Hospitals in Coimbatore. It contains around 6000 to 8000 rows of patient breath data for each recite, dating back to 2020. Tidal volume, inspiration and expiration ratios, reduced pressure, and PEEP pressure record were included in each row.

### 3.1 Realization of data

The Z-score method is a widely used statistical technique for data analysis. It helps understand how far a information point is from the average score. Z value of zero denotes that the data point's value is the identical as the average score of the data set. On the other hand, a Z-score of one signifies a value that is a single variation from the norm. A positive Z value implies that the information point's score is greater than the average, while a negative Z-score indicates that it is less extreme than the mean. Using the Z-score technique, researchers can easily compare and interpret data points in a standardized way.

### 3.2 Single hot encoding

Single hot encoding is a method employed in neural network to modify categorical data into numerical data. This technique transforms categorical-type factors into binary traits that are heated encoding. The idea behind this technique is that the components indicated by the encoded column acquire the value one, while the others receive a zero. In a particular effort, pressure and mechanical ventilation were transformed into categorical variables, where pressure was categorized as with and without a breathing apparatus.

### 3.3 Classifier methods

Fig 3 depicts the artificial intelligence algorithm and their categories. One subset of artificial intelligence is machine learning; artificial intelligence mirror the mind real neural connection. Machine learning is divided into several groups.

 Because of the focus on causative relationships and individual liability in medication, decision method are periodically used in models where the relevance of characteristics may support their use. Boosting is a strategy for improving performance that involves constantly replacing the old learner with the current learner. When one organization develops neural network models or with few examples, biased models may result. Exterior confirmation must be performed to assess the model's preciseness using data not used to develop the model. The accuracy of the neural network structure may be used to measure its generalizability. The prediction models were trained using grid search and 5-fold cross-validation approaches to generate models with the optimum excessive constraints.

 The analysis leverages mathematical modelling techniques to simulate the functioning of a breathing machine, and captures data using the transfer function. Pressure readings are

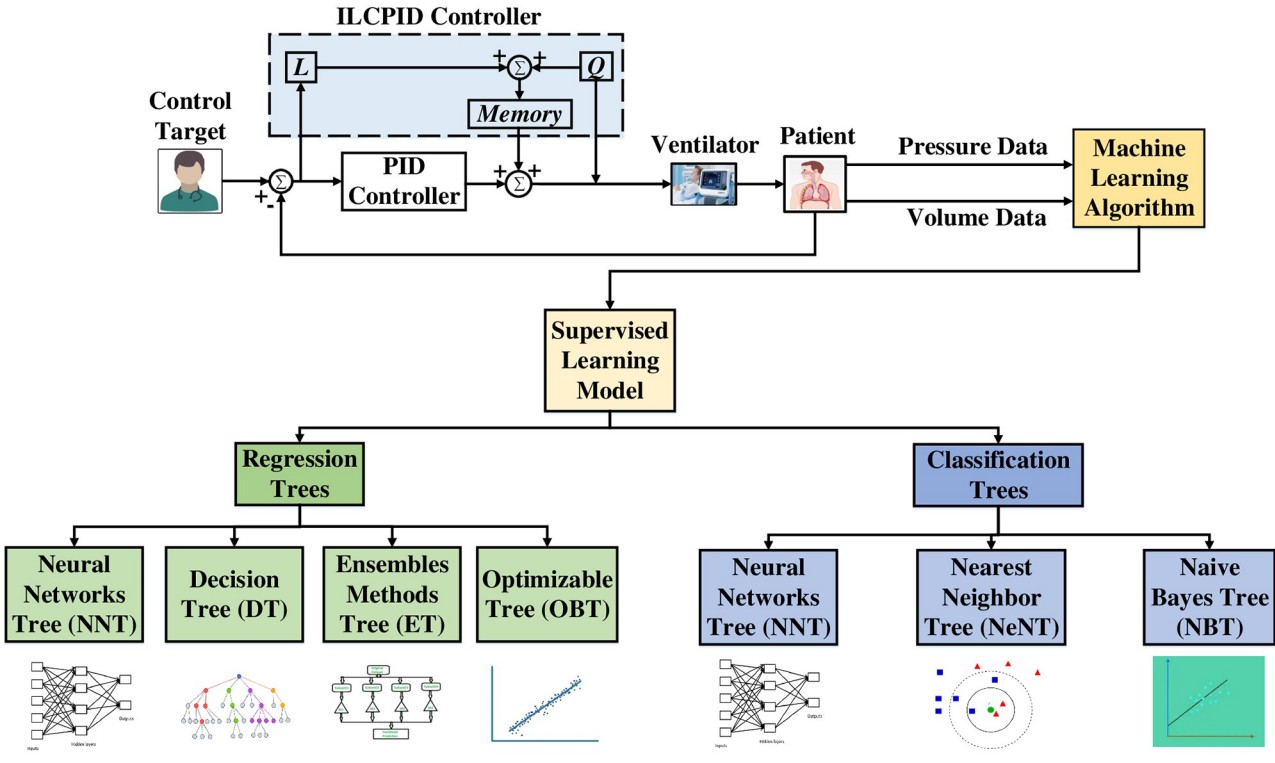

**Fig 3. Block diagram of ILC-PID based machine learning.**

recorded at 20 cm of $H_2O$, and then increased to 40 cm of $H_2O$ using the ILC. Data preprocessing is a fundamental process in various data-related tasks aiming to enhance the efficacy and efficiency of data analysis through standardization and normalization. This process involves organizing and transforming data to ensure consistency in data representation, enabling comparison and analysis. The one-hot encoding technique represents categorical variables as binary vectors. At the same time, numerical format conversion is utilized to transform categorical information into a numerical structure for analysis or model training purposes. In this research, the data preprocessing steps involve importing the dataset, importing libraries, and partitioning the dataset into training (70%) and testing (30%) sets. Further, feature derivation is a critical component of data preprocessing, involving selecting, extracting, and transforming relevant features from the available data to build more accurate and efficient machine-learning models. Model selection is based on performance metrics such as precision, sensitivity, training time, and accuracy. In machine learning, model training involves inputting an ML algorithm to enable it to recognize and learn optimal values for relevant attributes. The initial stage in machine learning model development is model training, yielding a functional model that can be validated, tested, and deployed. Data processing to prepare data for model construction, algorithm selection, model construction, performance metric computation, and selection of the better-performing model are all critical components of ML model development. Fig 4 shows the overall flow of the proposed technique and the developed classification framework using different phases. This task's solution incorporates a number of well-known approaches from the area, such as decision model, optimized algorithms, naïve, closest neighbour, ensembled, and neural model. The suggested approach will be highly effective in extracting meaningful information from complex data.

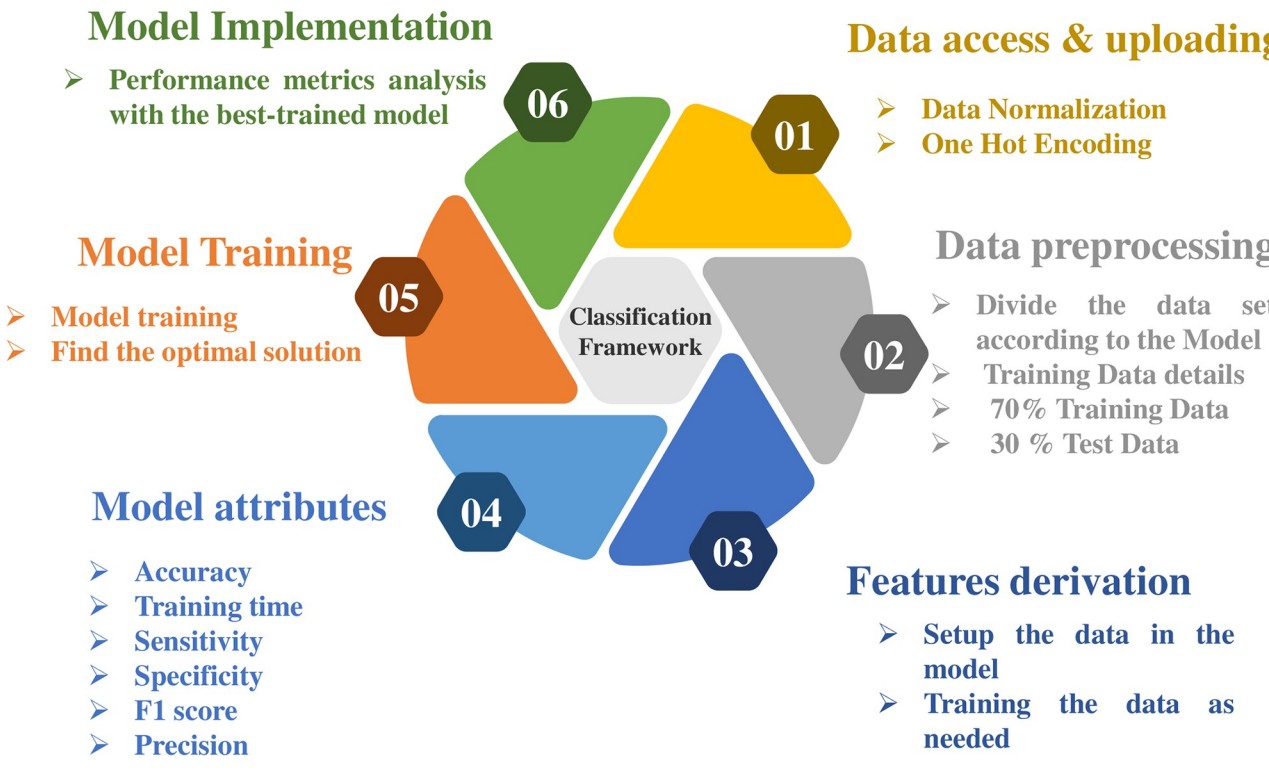

**Fig 4. Overall flow chart of developed classification framework.**

### 3.4 Cross-validation strategy

To evaluate the network, we used the k-fold cross-validation approach. We randomly divided the dataset into k incompatible clusters and used k-1 of these clusters to educate the group while keeping one subset for testing. We repeated these stages until all k subgroups had been examined. Every program period used 70% of the instruction set, and the remaining was kept as the verification set to ascertain if the ventilator model gained greater training accuracy. In this work, we place k to 5 folds. Fig 5 illustrates the precise operational steps.

**3.4.1 Classifier based on a neural network.** Understanding neural network models is complex, and model adaptability is increased by the artificial neural network's entire layer count and size. Every model is completely connected feed-forward neural network for classification. The first fully correlated neural network layer is connected to the one preceding it. Each layer that is ultimately interconnected doubles the data using a weight matrix, which is carried by a biasing function. The network output is generated by the last eventually coupled layer and the following activation function: anticipated categorization and classification scores. The neural network's representational capacity and number of layers are simply adjustable. It is a model of parametric neural prediction that is complex. In which case $I_1, \ldots I_7$ is the input node, $O_1, O_2$ is the output node, and $H_1, H_2, H_3$ is the hidden node.

1. Establish sets of shared activation functions for each hidden node. $H_i$ for

$$\text{for} \begin{cases} H_1: & (-1, 0, 1) \\ H_2: & (0, 1) \\ H_3: & (-1, 0.24, 1) \end{cases}$$

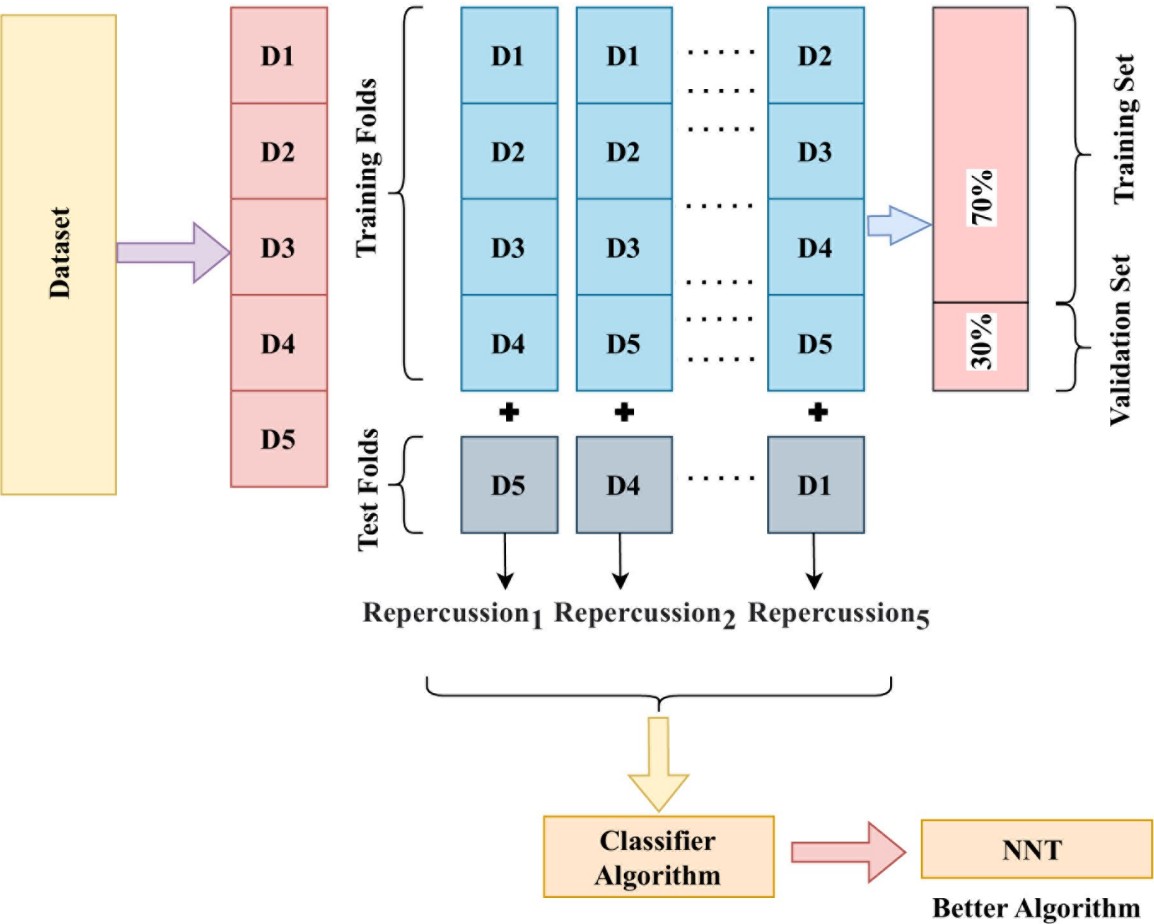

**Fig 5. Five-fold cross-validation strategy.**

2. Create rules that link output nodes $O_j$ that have comparable activation functions
   IF$(H_2 = 0 AND H_3 = -1)$ *OR*
   $(H_1 = -1$ AND $H_2 = 1$ AND $H_3 = -1)$ *OR*
   $(H_1 = -1$ AND $H_2 = 0$ AND $H_3 = 0.24)$
   THEN $O_1 = 1, O_2 = 0$
   ELSE$O_1 = 0, O_2 = 1$

3. Input nodes $I_j$ and output nodes $O_j$ association guidelines ought to be established
   IF $(I_2 = 0$ AND $I_7 = 0)$ THEN $H_2 = 0$
   IF$(I_4 = 1$ AND $I_6 = 1)$
   THEN $H_3 = -1$ IF$(I_5 = 0)$ THEN $H_3 = -1$

4. Provide rules for the input and output categorization
   IF$(I_2 = 0$ AND$I_7 = 0$ AND$I_4 = 1$ AND $I_6 = 1))$ THEN class = 0
   IF$(I_2 = 0$ AND$I_7 = 0$ AND$I_5 = 0)$THEN class = 1

   **3.4.2 Classifier.** A classifier is an algorithm that is trained to categorize incoming data into specific classes. Its goal is to identify patterns and correlations in the training set and determine a mapping involving the input feature sets and the output categories [3].

**3.4.3 Decision based classifier.** A classifier that bases judgments on a set of rules or criteria obtained from the training data is referred to as a decision category in general. To choose the best class or category for a particular instance, the input features are compared to these rules as part of the decision-making process [6].

**3.4.4 Optimizable bayes.** In order to practice Bayesian optimization Bayesian optimization uses objective function evaluation and upholds an internal Stochastic process framework for the goal function. One development in the approach employs an acquiring mechanism for Bayesian optimization to assess the following points [33].

**3.4.5 Naive bayes.** The naive Bayes categorization is a functional Bayesian evaluation approach whereby a novel occurrence is supplied together using a number of objective function scenarios for training represented by the tuple of characteristics $\langle a_1, a_2, \ldots, a_n \rangle$. The learner must predict the objective worth of this current instance. The new model is categorized using the Bayesian approach by deciding on the best probable objective, $v_{MAP}$, considering the feature values $\langle a_1, a_2, \ldots, a_n \rangle$ that define the instance [34].

**3.4.6 Confusion matrix.** The confusion matrix identifies the features of a categorization law. It provides the quantity of parts classified properly or poorly for each group. The prominent diagonal shows number of events were incorrectly categorized within each group. The off-diagonal components represent the quantity of remarks that were incorrectly classified. People compare the actual class assigned to each instant in the test set to the one given by the trained classifier [3].

**3.4.7 Ensemble trees.** Instead of typical methods of learning that try to create a single learner out of training data, the ensemble methodology develops and combines a collection of learners. Learners in this context are the individuals who comprise an ensemble. Base learners are often created by a decision model, artificial neural model, or alternative learning algorithm as the foundational learning algorithm using suggested training data [35].

## 3.5 Metrics for classification

The following equations expresses the formula for finding metrics for classifier algorithm.

**Accuracy.** The percentage of correctly identified items determines accuracy.

$$\text{Accuracy} = \frac{t_p + t_n}{t_p + t_n + f_p + f_n} \tag{4}$$

where $t_p$—True Positive, $t_n$ -True Negative, $f_p$ -False Positive, $f_n$—False Negative.

**Precision.** Precision uses the same basic idea as recall, but the emphasis is now on False Negatives rather than False Positives. Once more, the Real Negatives are not taken into account.

$$\text{Precision} = \frac{t_p}{t_p + f_p} \tag{5}$$

**Sensitivity.** The number of positive results our ML model returned can be called recall. With the aid of the following formula, designers can quickly calculate it using the confusion matrix.

$$\text{Sensitivity} = \frac{t_p}{t_p + f_n} \tag{6}$$

**Specificity.** The number of negatives our ML model returned can be considered specificity, which contrasts with recall.

$$\text{Specificity} = \frac{t_n}{t_n + f_p} \tag{7}$$

**F1 score.** The F1 score is calculated as the balanced average of recall and accuracy. The F1 score will yield a value ranging from zero to one. A score of a value of on the F1 measure indicates perfect recall and precision. The F1 score is zero if neither the recall nor the precision are zero.

$$F1 = 2 * (precision * recall)/(precision + recall) \tag{8}$$

## 4 Iterative learning Controller based PID controller

Iterative Learning Control (ILC) is an effective method for increasing system outcomes through ongoing instruction obtained from reiterated system interactions, much like how people learn new skills through doing tasks repeatedly, such as driving a motorcycle, automobile, which comes from ongoing instruction and learning experiences [36–38]. The ILC even with unpredictable models, the controller lowers transient tracking error and improves system efficiency by assimilating error information for each cycle [39, 40].

It is worth mentioning that the ILC approach controls the control signal's input levels, while alternative learning with adaptation controls systems change entirely the connected processors. ILC is a helpful instrument for industrial time-delayed one and repeated action applications because of the performance increase acquired via repeated learning practices [39, 41, 42]. In uncertain conditions, including classical controllers the feed-forward route is a popular method in the design of closed-loop management framework with single-input, single-output designs. ILC is distinct from other control systems, such as adaptive model, neural networks, and repetitive method. As a result, by integrating the PID as a feedback controller with the ILC, this study creates a present cyclic feedback ILC-PID controller.

The typical PID controller's control signal is created by combining the error signal $E(s)$, the process output $Y(s)$, and the parameters proportional gain ($K_p$), integral gain ($K_i$), and derivative gain ($K_d$). Its equation is as subsequently:

$$U(s) = \left( K_p + \frac{K_i}{Ts} + K_d s \right) E(s), \ E(s) = R(s) - Y(s). \tag{9}$$

To derive the learning rule for the ILC controller setup, the proper function information, namely the Q filter, L filter, and controller (C), must be used.

$$u_{i+1} = Q u_i + L e_i + C e_{i+1}. \tag{10}$$

The cut-off frequency of the ventilator transfer function and the frequency response of the bode plot will be used to calculate the L and Q filters in the ILC controller. In the Laplace domain, ultimate analogous controller formula is produced by stating them in form of input-output relationship, making the ILC-PID formula as subsequently:

$$\bar{u}(s) = (1 - Q)^{-1} \left( L + \left( K_p + \frac{K_i}{s} + K_d s \right) E(s) - U(s) \right) \bar{E}(s) \tag{11}$$

Fig 6 shows the revised structure that can be adopted for the existing cyclical feedback approach ILC-PID controller in a closed loop breathing apparatus using the aforementioned Eq 11.

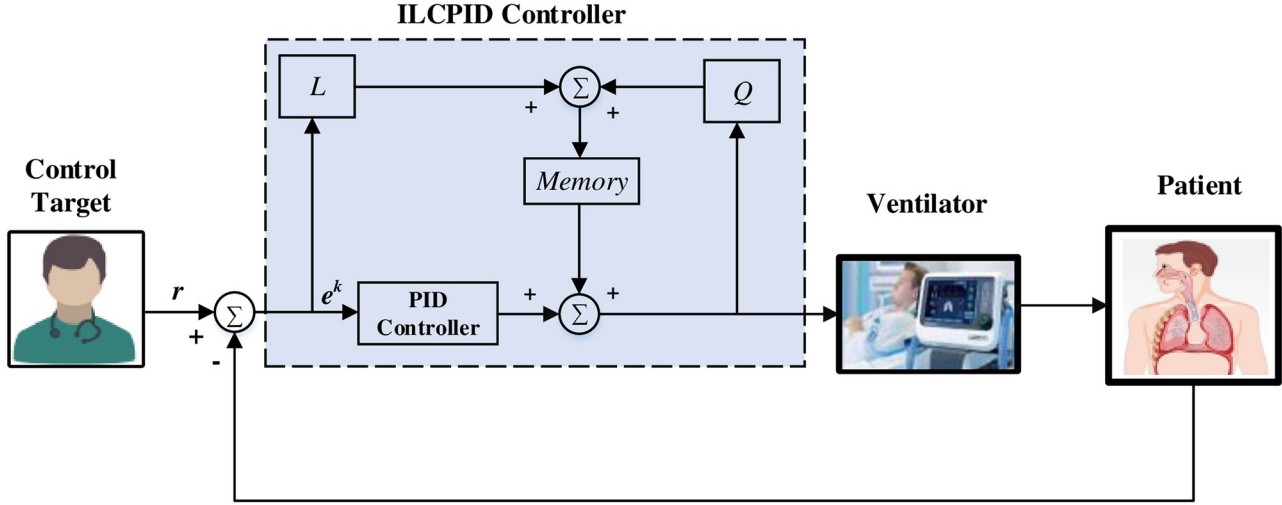

**Fig 6. Block diagram of ILC-PID controller.**

## 5 Results and discussion

This study compares the implementation of six different machine learning algorithms when imparted and evaluated on similar information. Considering the information. acquired, the goal is to determine the best performing methods for developing a enhanced accuracy throughout various airflow settings. In addition, this study uses model a real life analysis to select the optimal model to use in the two modes of Continuous Positive Airway Pressure (CPAP) and Proportional Assist Ventilation (PAV). The process of selecting a machine learning based classifier involves careful consideration of several hyper-parameters. The Gini diversity index is a crucial parameter to be considered for tree-based classifiers, while Bayes classifiers require using Bayesian optimizers. Nearest neighbours classifiers rely on Euclidean distance, while ensemble classifiers such as AdaBoost are commonly used to improve the performance of individual classifiers. Finally, neural networks optimize performance by optimizing activation functions such as ReLU. Furthermore, classification techniques such as decision (DT), optimizable (OBT), naïve Bayes (NBT), closest neighbour (NeNT), ensemble classifier (ET), and neural network (NT) trees are used to select the optimum model for the ventilator. The measures utilized to evaluate and examine the effectiveness of machine learning models are F1 score, sensitivity, specificity, training length, accuracy, and F1 score. Their functionality is given in Tables 3 and 5 for various volumes and pressures of $H_2O$ at 20 cm. The findings and discussion of the ILC-PID controller are covered in the first half of this section. Second, the

**Table 3. Performance of volume ventilator system machine learning models.**

| Trees | Accuracy | | Training time | | Sensitivity | | Specificity | | F1Score | | Precision | |
|---|---|---|---|---|---|---|---|---|---|---|---|---|
| | CPAP | PAV | CPAP | PAV | CPAP | PAV | CPAP | PAV | CPAP | PAV | CPAP | PAV |
| **DT** | 69.2 | 70.32 | **0.84** | **0.7** | 0.581 | 0.643 | 0.532 | 0.772 | 0.365 | 0.421 | 0.775 | 0.863 |
| **OBT** | 68.23 | 76.31 | 36.25 | 28.7 | **0.729** | 0.841 | 0.625 | 0.752 | 0.623 | 0.741 | 0.628 | 0.823 |
| **NBT** | 72.95 | 82.3 | 79.27 | 75.15 | 0.658 | 0.771 | 0.628 | 0.756 | 0.632 | 0.831 | 0.689 | 0.771 |
| **NeNT** | 76.3 | 80.12 | 3.269 | 1.789 | 0.643 | 0.745 | **0.779** | **0.861** | 0.421 | 0.563 | 0.756 | **0.883** |
| **ET** | 74.2 | 80.12 | 4.568 | 3.85 | 0.685 | 0.852 | 0.329 | 0.445 | 0.521 | 0.632 | 0.721 | 0.745 |
| **NNT** | **81.6** | **84.59** | 4.563 | 3.96 | 0.705 | **0.861** | 0.778 | 0.812 | **0.775** | **0.845** | **0.812** | 0.823 |

machine learning interpretation for the volume is shown, proceeded by the findings and pressure respond by using confusion matrix calculation.

## 5.1 Performance of ILC-PID for the ventilator model

Figs 7 and 8 demonstrate the performance response of the PID and ILC-PID controllers for pressure. The findings demonstrate that without any intervention, the pressure climbs to 75 cm of $H_2O$ or psi of water within the specified duration. This in effect shows that the ILC-PID controller effectively helps to achieve the necessary high pressure, which helps to keep the breathing equipment that generates the pressure required for breathing in humans.

In the example above, the conventional controller can compress the maximum pressure of 40cm of $H_2O$. Figs 9 and 10 demonstrate the volume achieved using the ILC-PID controller and how it improves the respiratory system's capacity to create more volume. The proposed ILC-PID increases the tidal volume to a maximal of 3.5 ml from 0 ml, whereas the PID controller only reaches 2.1 ml.

## 5.2 Volume: Machine learning based classifier analysis

In the CPAP mode, Table 3 and Fig 11 provides a numerical assessment of the machine learning based classifier's performance at 20 cm $H_2O$ of volume. The correlation between the performance parameters is illustrated in the heat-map representation, which is given in Figs 12 and 13. From the Table 3 and Figs 14 and 15, the decision model achieves an average accuracy of 80.06%. A mean of 82.65% is obtained using the closest KNN classifier method, so the decision tree's preciseness elevates by 6.8% contrasted to the optimized model. The ensembled model accuracy was judged to be 69.5 percent. Eventually, the suggested neural method is the better-performing classifier compared to the remaining classifiers, with a Mean accuracy rate of 88.2%.

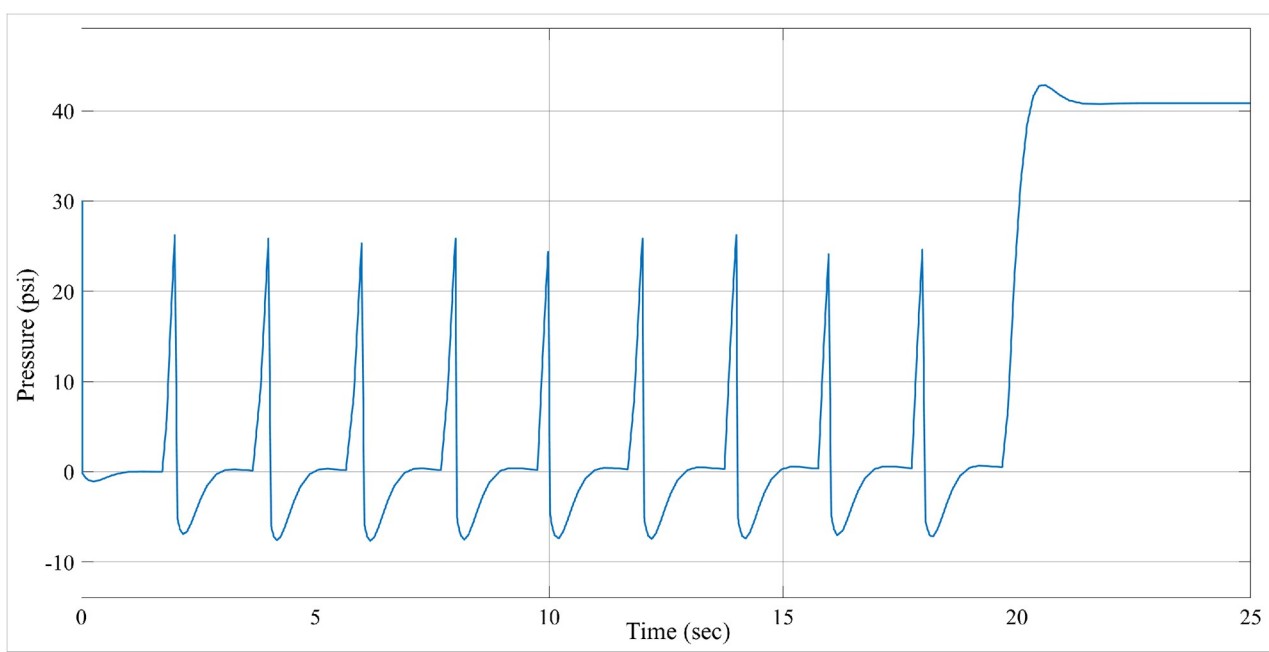

**Fig 7. Pressure increasing scenario for PID controller.**

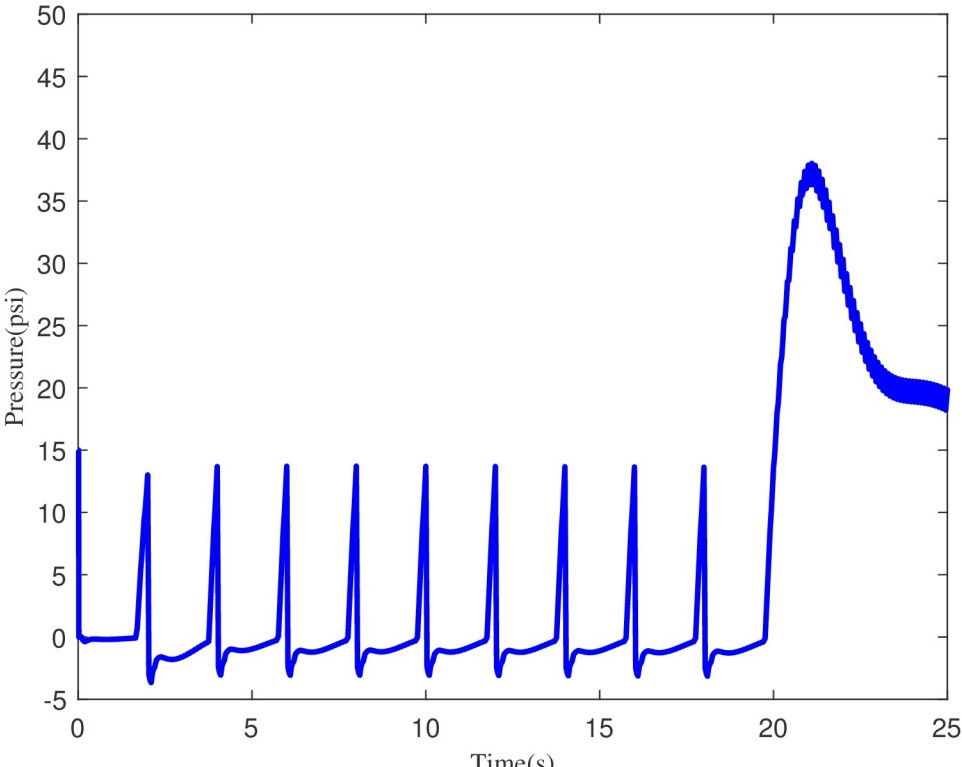

**Fig 8. Pressure increasing scenario for ILC-PID controller.**

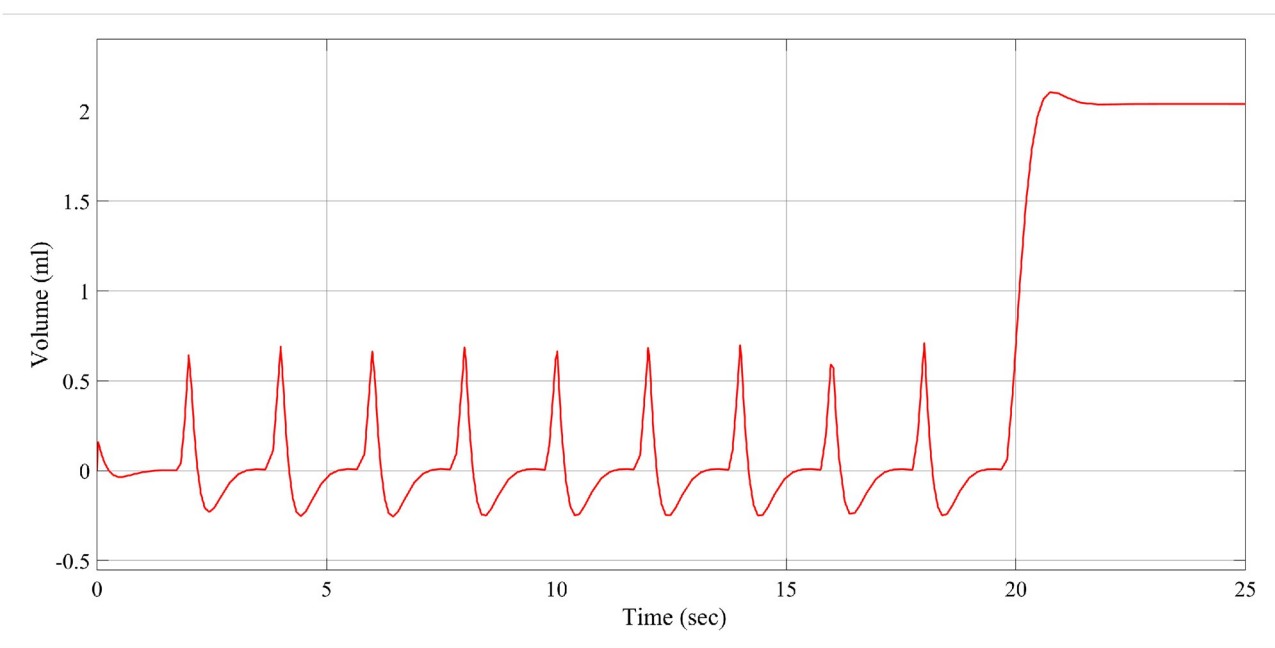

**Fig 9. Volume increasing scenario for PID controller.**

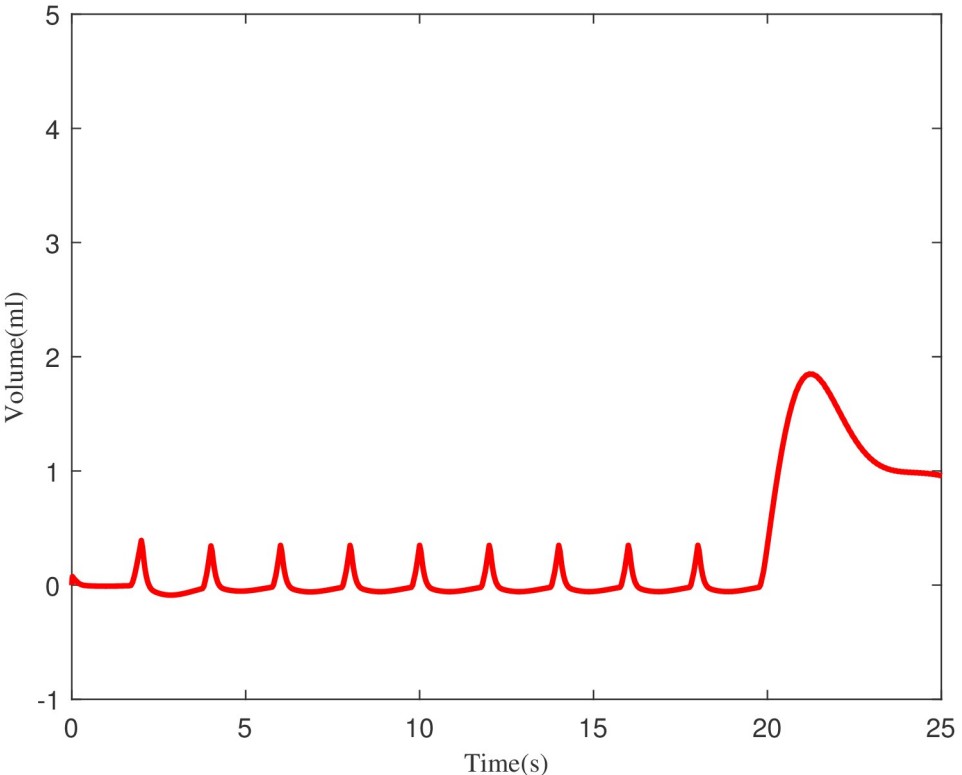

**Fig 10. Volume increasing scenario for ILC-PID controller.**

Table 4 shows a confusion matrix for both CPAP and PAV modes of a ventilator system employing volume control. Table 4 shows the determined analysis using the performance metrics equations. The confusion matrix containing various classifier trees of CPAP mode for volume is shown in Fig 14(a) to 14(f), respectively. It is clear that, utilizing results from and the confusion matrix figures, the accuracy was validated. In comparison to other classifiers, optimizable bayes and naive bayes have substantially longer training times (45.23 s and 89.5 s, respectively) and slower data processing rates. Regarding training time, the neural network tree takes 8.564 seconds to create, the ensemble tree takes 5.632 seconds, and the closest neighbour tree takes 3.78 seconds. The decision tree has the shortest training time of -1.268 seconds compared to other trees. The decision tree has a high sensitivity of 0.652%. The decision method accuracy rises by 0.14% contrasted to the optimized Bayes tree and by 0.823% with a mean of 0.824% is produced using the closest neighbour model. The ensemble model has a sensitivity of 0.742%. It will be more sensitive than while the tree was being monitored, as was already mentioned.

On average, the better closest neighbour classification tree in this has a preciseness of 0.824%. The decision model has a specificity of 0.623% on average. While the KNN classifier tree reaches a mean of 0.733%, the naive method decreases specificity by 0.562%, while the optimizable tree enhances specificity by 0.623%. The ensemble model is 0.118%. The top neural categorization has a mean accuracy rating of 0.524%. The decision method average F1 value is 0.5632. As the closest KNN classifier tree gains a mean of 0.821%, the F1 value of the naive Bayes model achieves by 0.632%. However, the F1 value of the decision tree decreases by 0.201% contrasted to the optimized model. The F1 value of the ensembled method is 0.723.

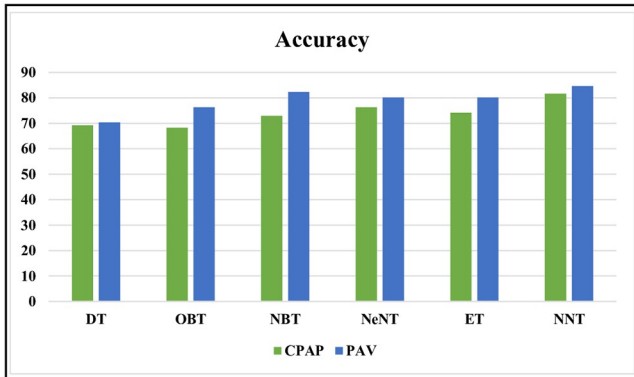

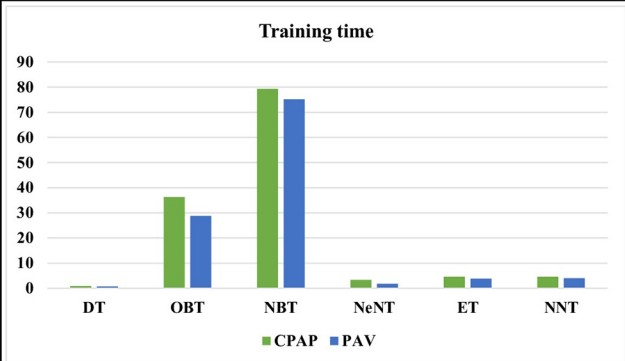

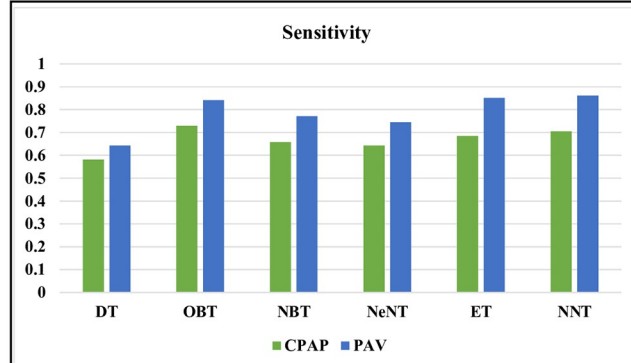

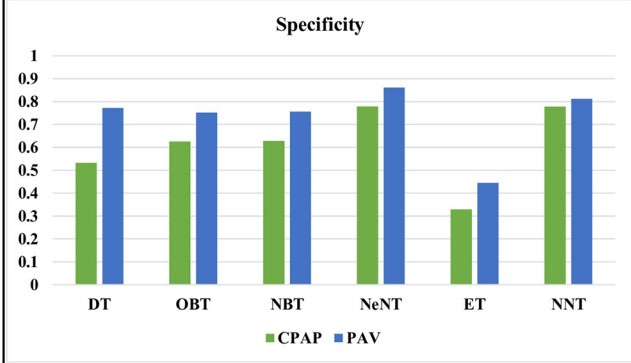

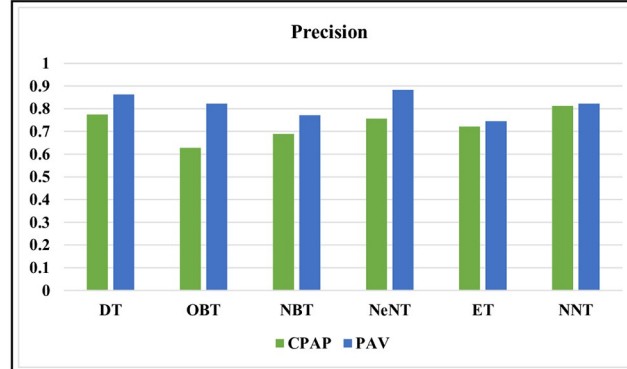

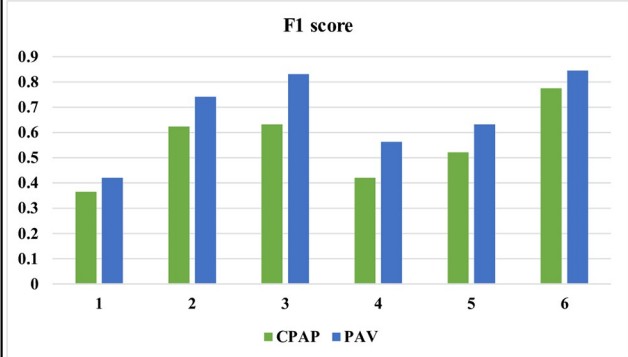

**Fig 11. Various performance metrics of volume.**

Finally, the better neural model has a mean precise rate of 0.689%. The preciseness of the decision model is approximately 0.635%. Contrasted to the optimized method, the naive model adds precision when the closest neighbour classifier tree averages 0.751%. However, the decision tree loses precision by 0.1915%. Ultimately, the proposed neural model had a mean precision of 0.854%.

In CPAV mode, the nearest neighbour model attains a mean of 82.65%., the decision tree's accuracy improves by 6.8% explored to the optimizable tree. Regarding training time, a neural network tree takes 8.564 seconds to form, an ensemble tree takes 5.632 seconds, and the closest

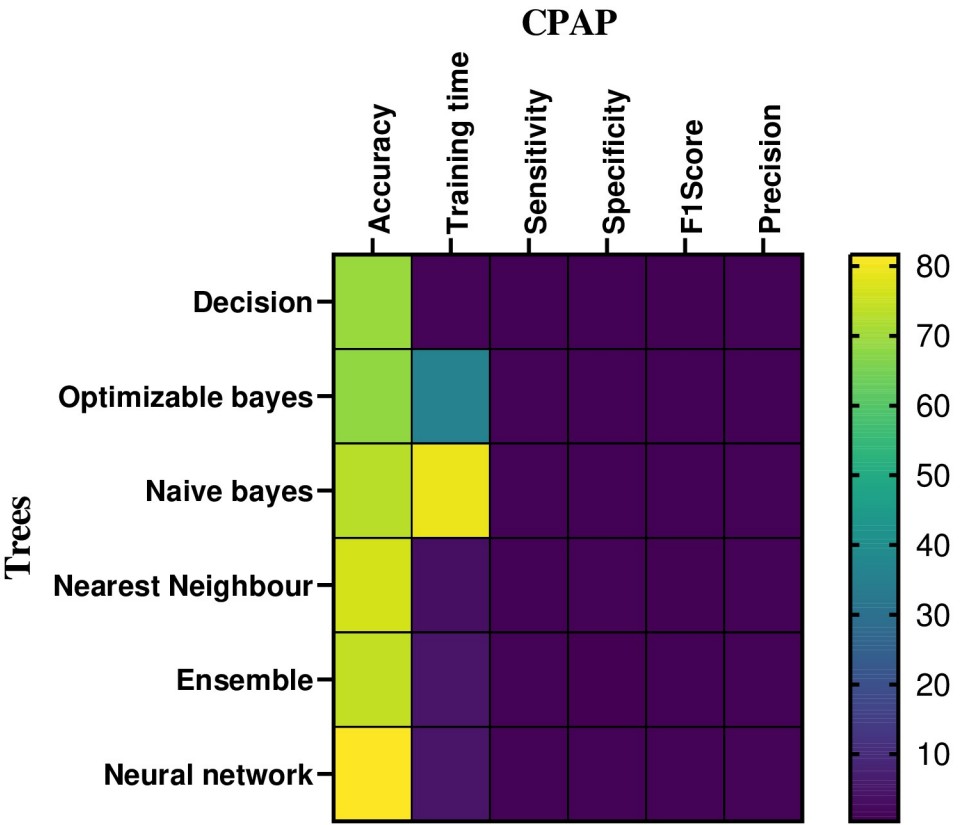

**Fig 12. Heat-map analysis of the CPAP for volume.**

neighbour tree takes 3.78 seconds. The sensibility of the ensemble classifier tree is 0.742%. It is now more sensitive than when the tree was examined earlier. When the closest neighbour classifier tree reaches a mean of 0.733% specificity, the naive Bayes model loses 0.562%, while the optimizable tree adds 0.623%. The ensembled categorization has a success rate of 0.118%. When the closest model reaches a mean of 0.821%, the naïve model F1 value rises by 0.632%, but the decision model's F1 value declines by 0.201% contrasted to the optimized method. The naive Bayes tree gains precision when the closest neighbour classifier tree averages 0.751%. However, the decision model loses precision by 0.1915% explored to the optimized model. The ensemble method has a precision of 0.629. Finally, the better neural model achieved a mean accuracy rate of 0.854%. The confusion matrix containing various classifier trees of PAV mode for volume is shown in Fig 15(a) to 15(f), respectively.

When both the naive Bayes and closest neighbour classifiers achieve a mean of 89.7%, the decision method accuracy is 6.2% greater than that of the optimized model. The ensemble model accuracy was judged to be 71.7%. Compared to other classifiers, optimizable and naive Bayes take much longer to train (28.7 and 84.06 seconds, respectively), and the data is also processed more slowly. Regarding training time, the neural network tree takes 4.0832 seconds, much longer than the 4.739 seconds required to build an ensemble tree or the 1.269 seconds needed to create a nearest method. The preciseness of the decision graph improves by 0.2007 percent in relation to the optimizer Bayes tree, with a median of 0.9225 percent obtained utilizing the closest neighbour classifier tree. The ensemble method has a sensibility of 0.902%. Specificity rises by 0.9167% when a mean of 0.7059% is generated

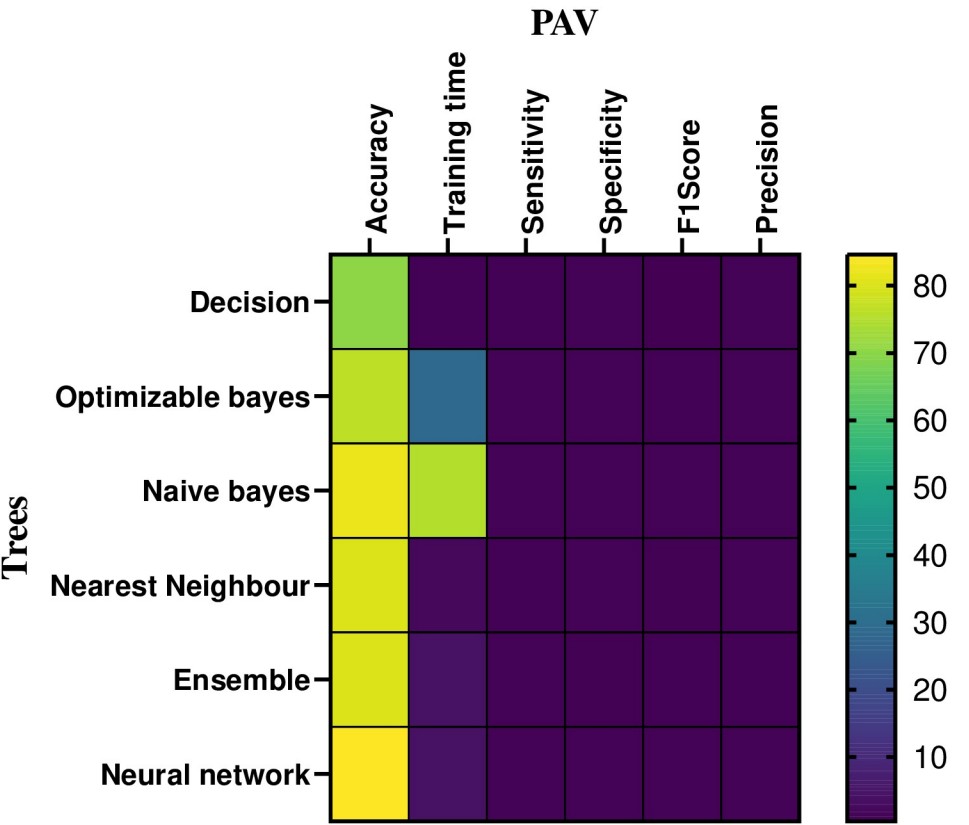

**Fig 13. Heat-map analysis of the PAV for volume.**

using most convenient neighbour classifier tree, whereas the naive model specificity falls by 0.7059%.

Contrasted to the optimized model the F1 value of the naive Bayes classification tree reduces by 0.004%, while the F1 value of the most nearby neighbour classification tree rises by 0.9402%. The ensemble method has an F1 value of 0.8178. Finally, the better neural method has a 0.9531% mean accuracy rate. While the closes neighbour classifier tree has a mean precision of 0.9593%, the decision tree has 0.08% less precision than the optimizable model. As mentioned earlier, the ensemble model has a higher specificity and accuracy than the another tree. Finally, the better optimizable network classification tree has a 0.9919% mean preciseness.

The performance of the mechanical ventilator in CPAP mode was evaluated using the ROC curve, as shown in Fig 16. This curve provides a comprehensive performance measure that takes into account all classification levels. The ROC curve can be evaluated as the probable that a model will rank a selected at arbitrarily positive example higher than a decided selected negative example. The values of ROC range between 0 and 1, where 0 indicates completely incorrect predictions and 1 represents perfect accuracy. In this study, the ROC curve revealed that the NBT technique had a lower precision value than ET in the mechanical ventilator system. The NNT technique was assigned an appropriate value on the curve. Notably, the neural network classifier demonstrated superior performance compared to all other techniques based on the ROC slope.

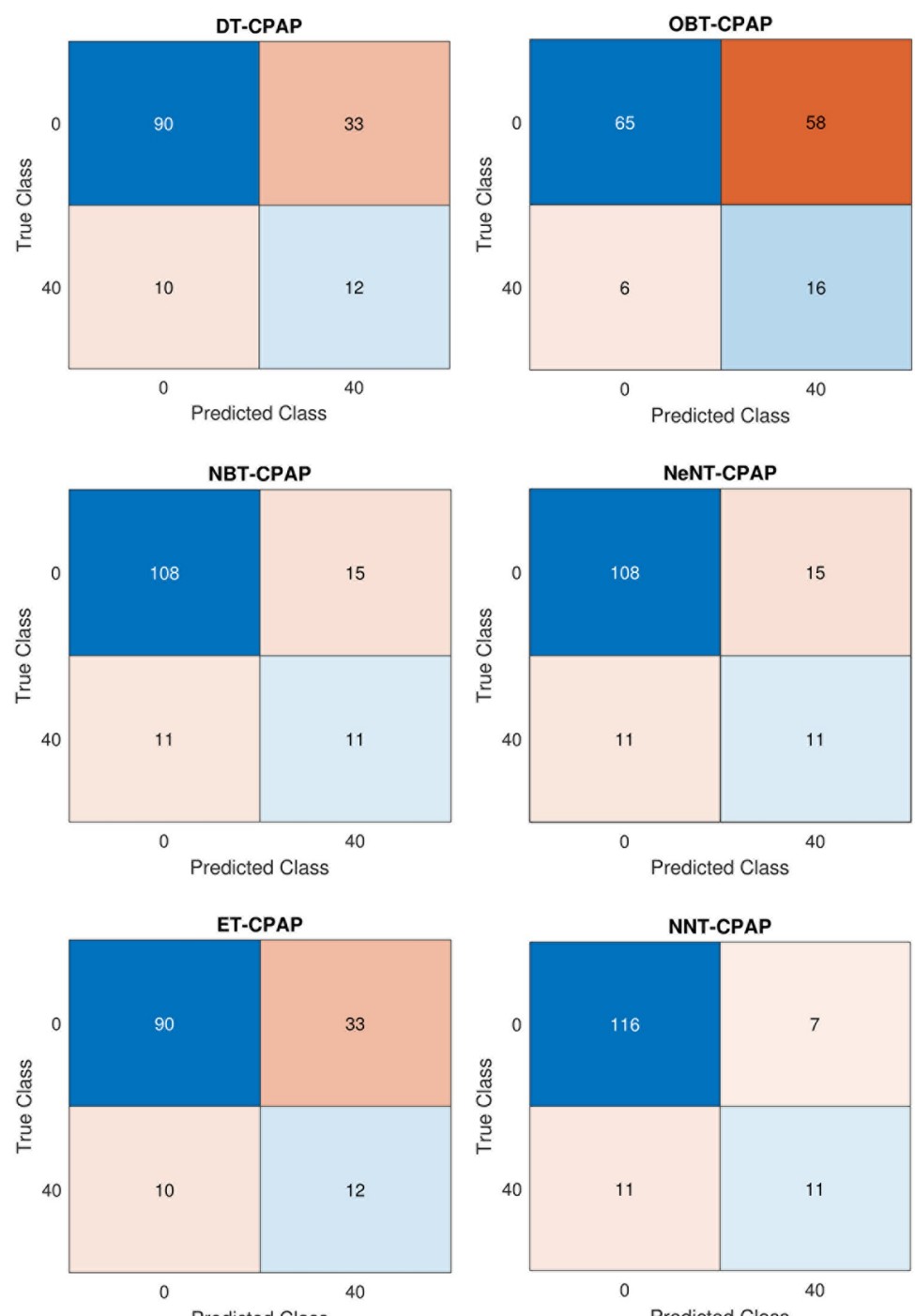

**Fig 14. Confusion matrix in CPAP Mode for Volume (a) DT (b) OBT (c) NBT (d) NeNT (e) ET (f) NNT.**

The ROC curve for the breathing apparatus performs well whenever utilized to volume in PAV mode, as shown in Fig 17. Compared to DT, the ROC curve for this ventilation system reveals a lower OBT value for accuracy. The NNT has been assigned an accurate value that correlates to the curve. The neural network classification tree outperformed earlier techniques by a significant margin in terms of the ROC curve.

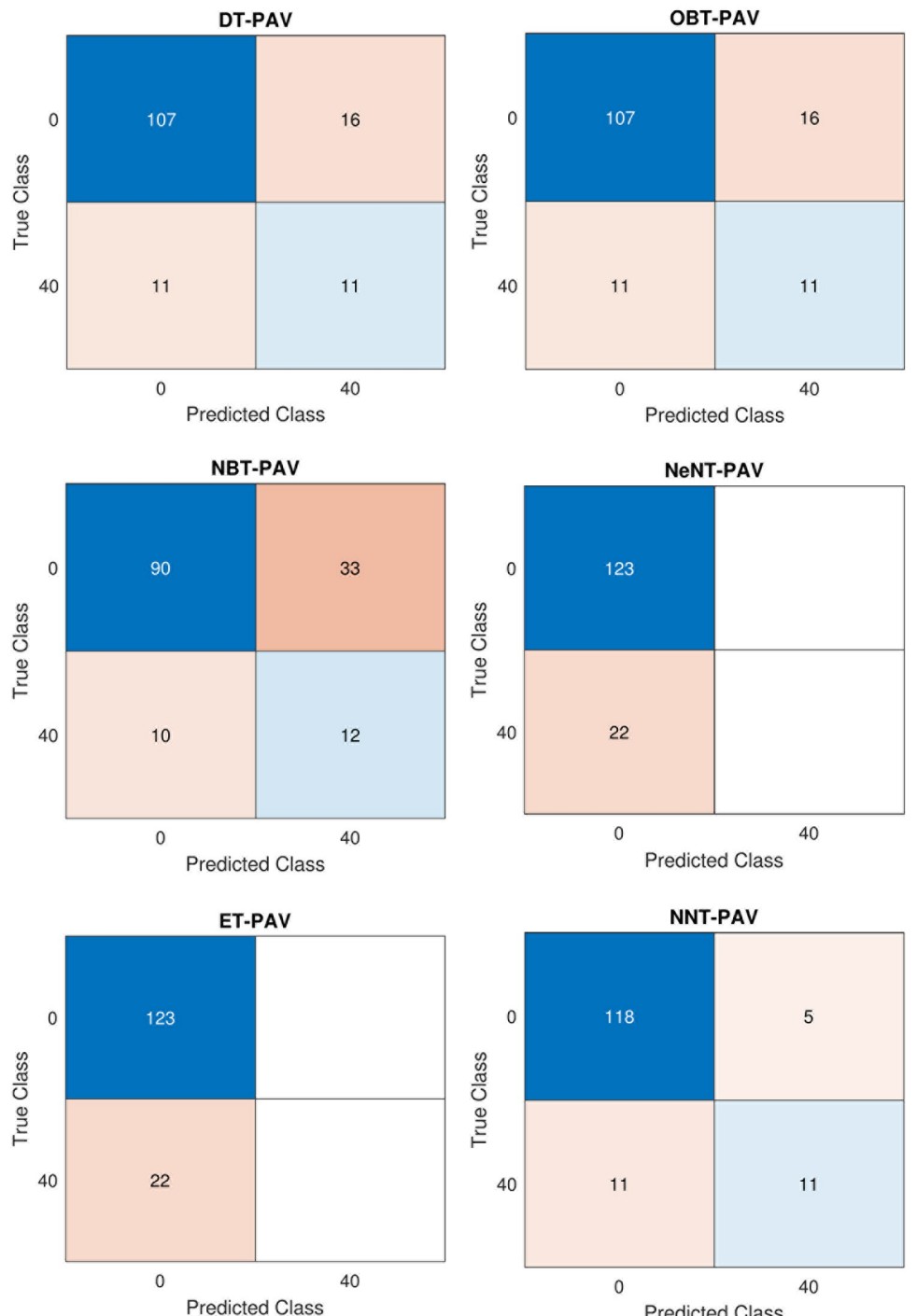

**Fig 15. Confusion Matrix in PAV Mode for Volume (a) DT (b) OBT (c) NBT (d) NeNT (e) ET (f) NNT.**

## 5.3 Pressure: Machine learning based classifier analysis

The pressure performance of the machine learning based classifier in CPAP and PAV modes is numerically assessed in Table 5 and Fig 18. Figs 19 and 20 shows the mechanical ventilator's pressure heat map depiction in CPAP and PAV modes. Based on the data, the decision model

**Table 4. Confusion Matrix Analysis of a ventilator system of volume.**

| Trees | CPAP | | | | PAV | | | |
|---|---|---|---|---|---|---|---|---|
| | $t_p$ | $t_n$ | $f_p$ | $f_n$ | $t_p$ | $t_n$ | $f_p$ | $f_n$ |
| DT | 90 | 12 | 33 | 40 | 107 | 11 | 16 | 11 |
| OBT | 65 | 16 | 58 | 6 | 107 | 11 | 16 | 11 |
| NBT | 108 | 11 | 15 | 11 | 90 | 12 | 33 | 10 |
| NeNT | 108 | 11 | 15 | 11 | 123 | 0 | 0 | 22 |
| ET | 90 | 12 | 33 | 10 | 123 | 0 | 0 | 22 |
| NNT | 116 | 11 | 7 | 11 | 118 | 11 | 5 | 11 |

has a mean accuracy of 86%. As the closest neighbour Classifier method gains a mean of 82.6%, the decision method preciseness rises by 6.8% explored to the optimizable tree.

The ensemble classifier has a 69.5% accuracy. Finally, the suggested neural model tree is the better-performing classifier, with a mean preciseness of 88.2%. Optimizable and naïve bayes had approximately more extended training periods (45.23s and 89.5s, respectively) and slower data processing rates than other classifiers. Regarding training time, a neural network tree takes 8.564 seconds, an ensemble tree takes 5.632 seconds, and the closest neighbour tree takes 3.78 seconds. The decision tree has the shortest training time of -1.268 seconds compared to other trees. Contrasted to the optimizable Bayes tree, the decision method accuracy rises by 0.14% and by 0.823% when a classifier's closest neighbour tree yields an average of 0.824 percent. The sensitivity of the ensembled model is 0.742%. As previously stated, it is more

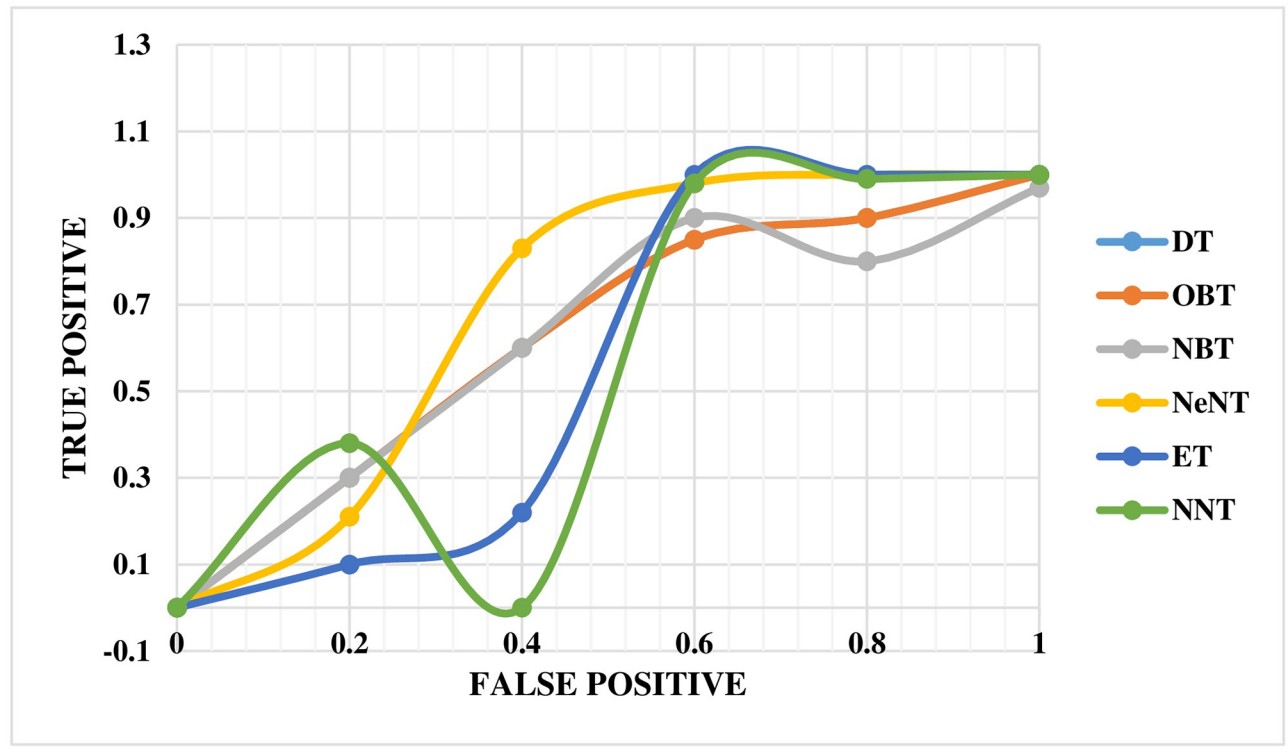

**Fig 16. ROC curve for volume in CPAP mode.**

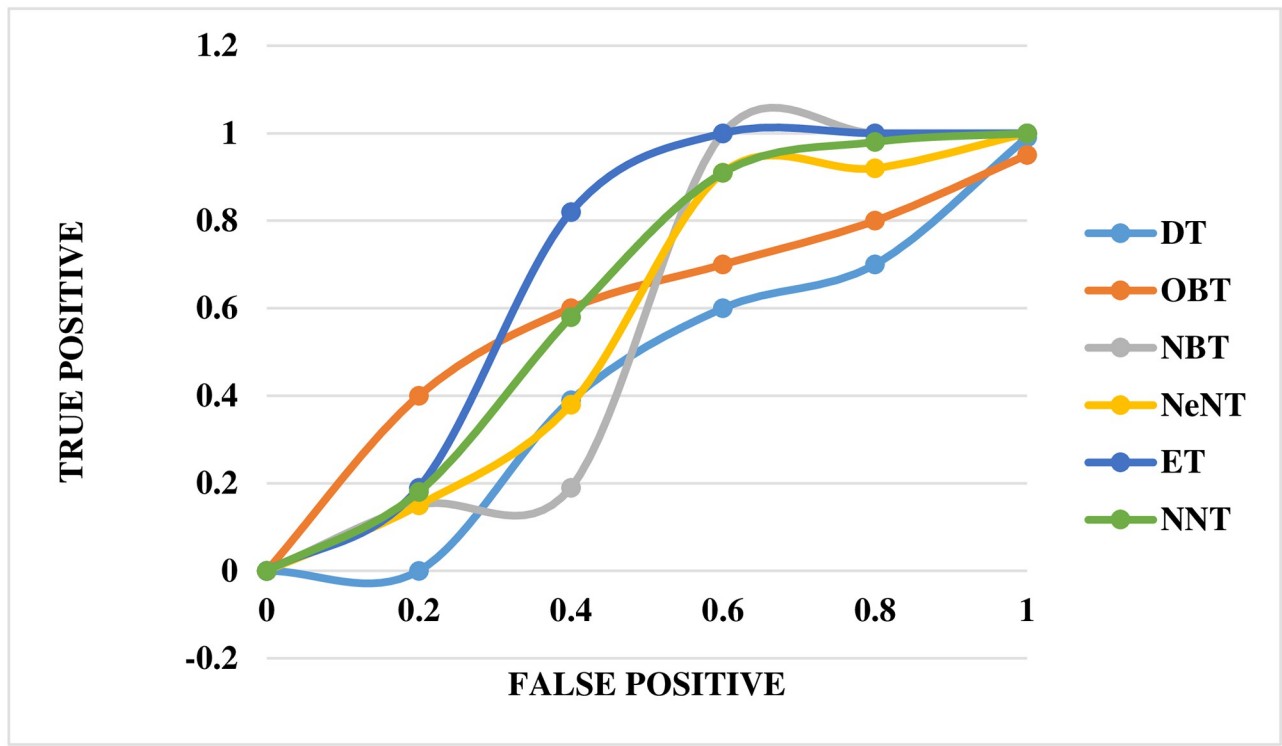

**Fig 17. ROC curve for volume in PAV mode.**

sensitive than it was while the tree was being studied. The better closest neighbour classification tree in this has a mean accuracy rate of 0.824%.

The decision method has an mean specificity of 0.623% based on Table 5. When the KNN classifier tree reaches a preciseness of 0.733% specificity, the naive Bayes model loses 0.562%, while the optimizable tree adds 0.623%. The ensemble classifier tree technique has a success rate of 0.118%. The top neural method has a mean accuracy rating of 0.524%. On average, the decision model has an F1 value of 0.5632%. When the closest neighbour model reaches a mean of 0.821%, the naive Bayes method F1 value rises by 0.632%, whereas the F1 value of the decision tree drops by 0.201 percent in comparison to the optimizable tree. The ensemble classifier method has an F1 value of 0.723. Ultimately, the superior neural network categorization structure has a correctness rate of 0.689% on average.

**Table 5. Performance of machine learning models of a ventilator system of pressure.**

| Trees | Accuracy | | Training time | | Sensitivity | | Specificity | | F1Score | | Precision | |
|---|---|---|---|---|---|---|---|---|---|---|---|---|
| | CPAP | PAV | CPAP | PAV | CPAP | PAV | CPAP | PAV | CPAP | PAV | CPAP | PAV |
| DT | 86 | 91 | **1.268** | 0.9248 | 0.652 | 0.7167 | 0.623 | 0.8462 | 0.5632 | 0.949 | 0.635 | 0.9837 |
| OBT | 79.2 | 84.8 | 45.23 | 28.7 | 0.823 | 0.9173 | **0.725** | **0.9167** | 0.765 | **0.9531** | **0.8265** | **0.9919** |
| NBT | 78.2 | 89.7 | 89.5 | 84.06 | 0.785 | **0.9225** | 0.562 | 0.7059 | 0.632 | 0.9487 | 0.628 | 0.9597 |
| NeNT | 79.3 | 89.7 | 3.78 | **1.269** | **0.824** | 0.9219 | 0.623 | 0.7059 | **0.821** | 0.9402 | 0.751 | 0.9593 |
| ET | 69.5 | 71.7 | 5.632 | 4.739 | 0.742 | 0.902 | 0.118 | 0.2791 | 0.723 | 0.8178 | 0.629 | 0.748 |
| NNT | **88.2** | **91.7** | 8.564 | 4.0832 | 0.695 | 0.9173 | 0.524 | **0.9167** | 0.689 | **0.9531** | 0.854 | **0.9919** |

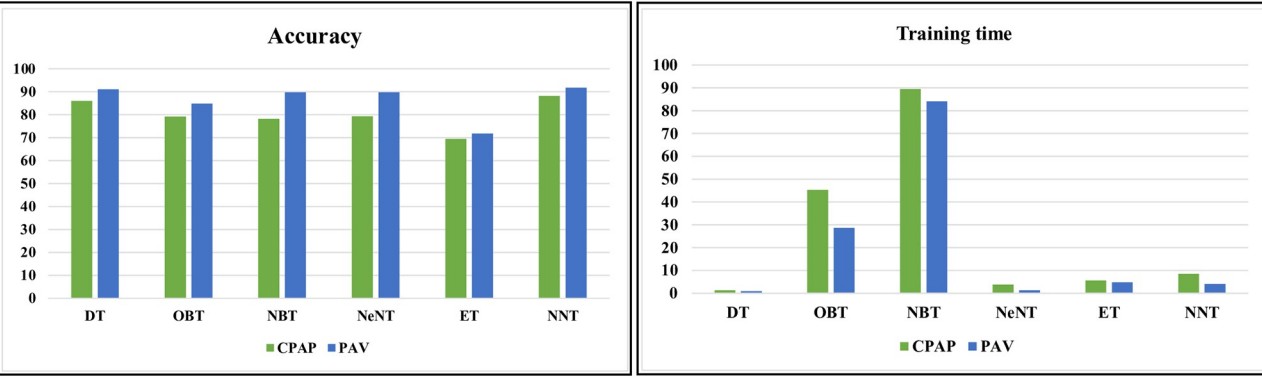

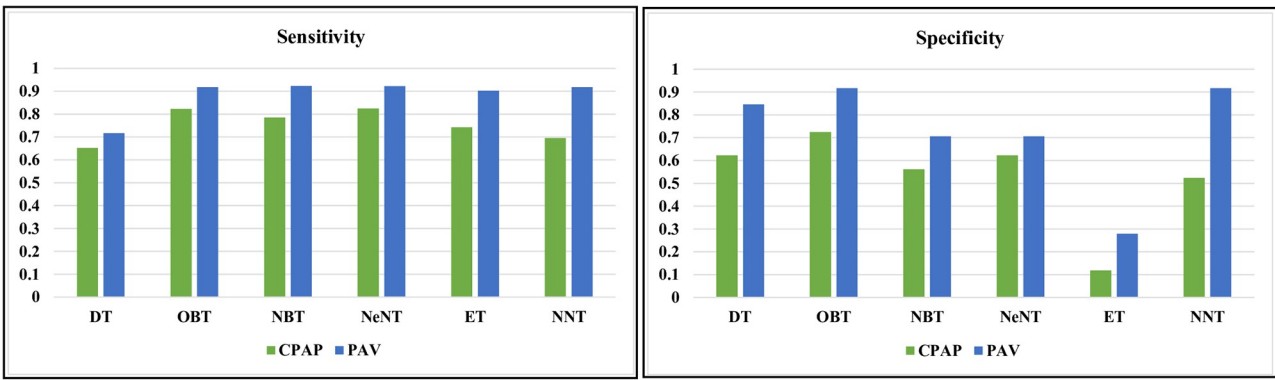

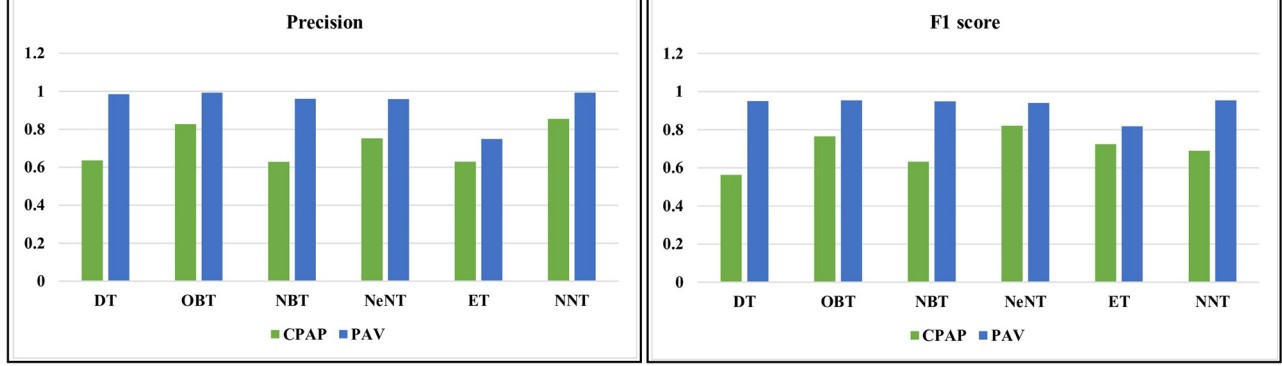

**Fig 18. Various performance metrics of pressure.**

Regarding classification algorithms, precision and accuracy are crucial factors to consider. The decision tree model typically achieves a precision rate of 0.635%. However, the naive Bayes tree and nearest neighbour classifier tree tend to outperform it, achieving an average precision rate of 0.751%. On the other hand, when explored to the optimizable tree, the decision tree model falls short, with a decrease in accuracy of 0.1915%. The ensemble categorization, on the other hand, has a precision rate of 0.629%. Taking all these factors into account, the neural network classification tree proves to be the better-performing classifier, with a mean accuracy of 0.854%. The decision tree, while not the better-performing, still has a

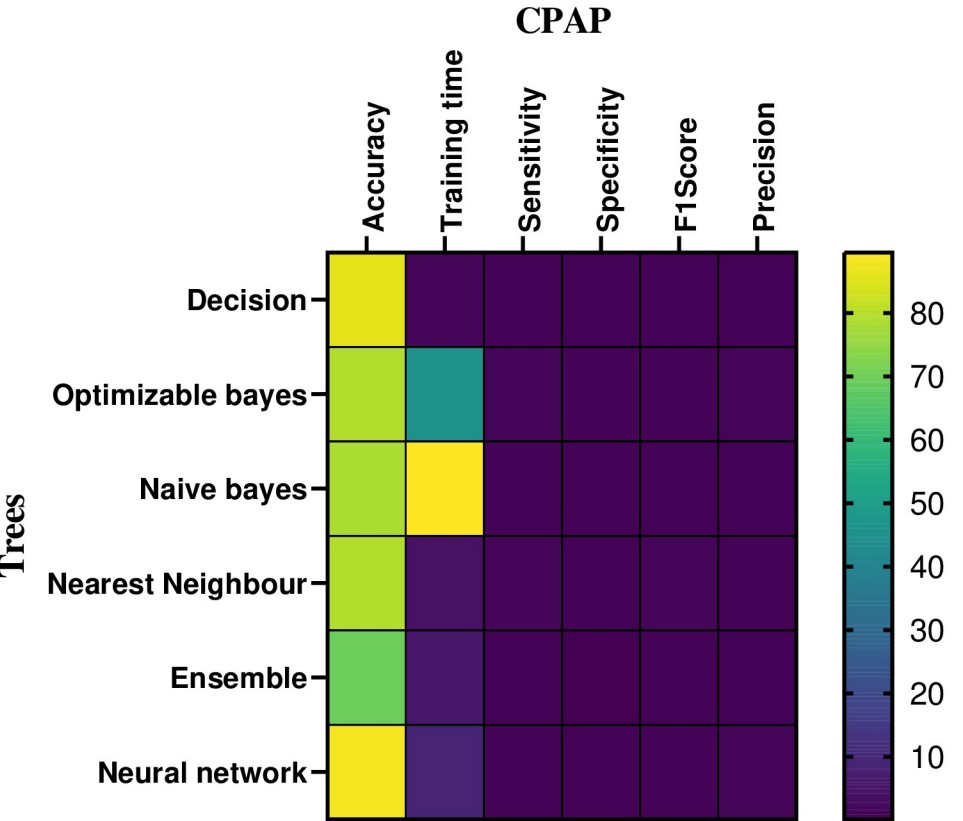

**Fig 19. Heat-map analysis of the CPAP for pressure.**

respectable average accuracy rate of 91%. It even outperforms the optimizable tree, achieving an accuracy rate that is 6.2% greater. Meanwhile, the naive Bayes and nearest neighbour classifiers perform similarly, achieving an average accuracy rate of 89.7%. Lastly, the ensemble classifier tree model must catch up with a 71.7% accuracy rate. The proposed neural network tree is the clear winner, with a mean of 91.7%. Table 6 shows a confusion matrix for both CPAP and PAV modes of a ventilator system employing pressure control. Fig 21(a) to 21(f) depict the confusion matrix comprising different classifier trees for CPAP mode for pressure.

Compared to other classifiers, optimizable and naive bayes take approximately longer to train (28.7 and 84.06 seconds, respectively), and the data is also processed slower. Regarding training time, the neural network tree takes 4.0832 seconds, approximately longer than the 4.739 seconds necessary to construct an ensemble tree or the 1.269 seconds required to prepare a closest neighbour tree. Compared to other trees, the nearest neighbour tree has the shortest training time of 1.269 seconds. The decision tree has a sensitivity of 0.7167% on average. As a mean preciseness of 0.9225% is gained employing the closest neighbour categorization, the decision model accuracy rises by 0.2007% explored to the optimized bayes model. The sensibility of the ensemble model 0.902%. As previously said, their senses are more accurate than when the tree was studied. The better closest neighbour classification tree„ in this case, gets an average accuracy rating of 0.9173%. Fig 22(a) to 22(f) depict the confusion matrix comprising different classifier trees for PAV mode for pressure.

Despite the naive categorization specificity reduces by 0.7059% when a mean of 0.7059% is produced employing the closest categorization, specificity rises by 0.9167%. The ensemble

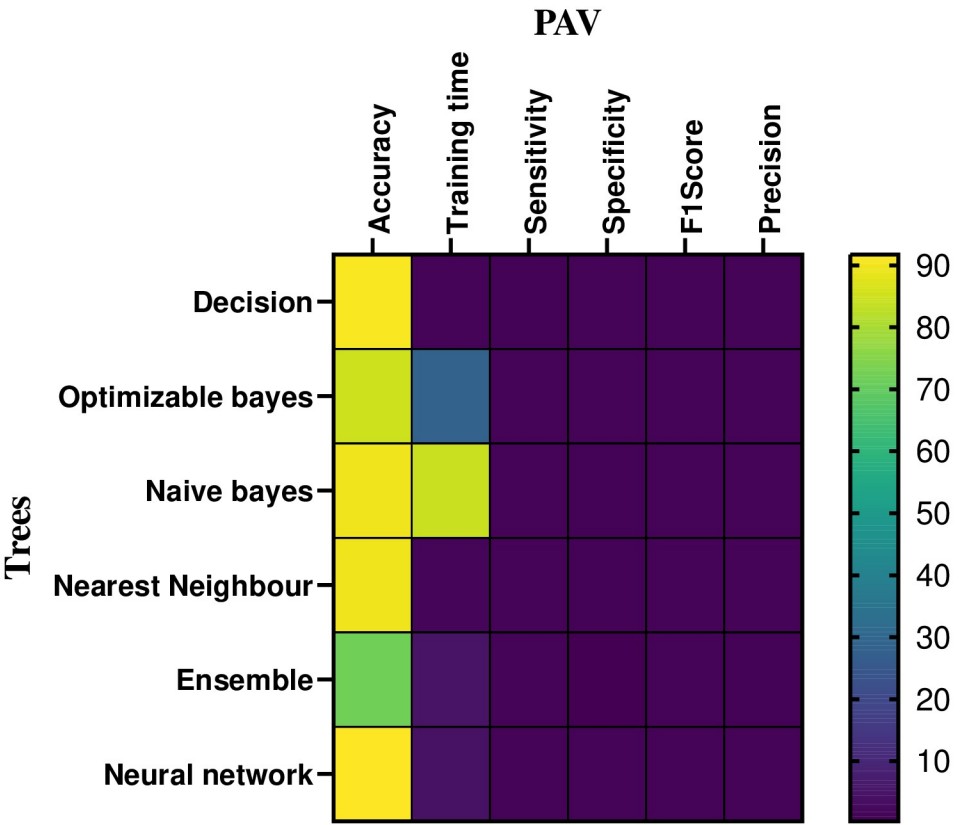

**Fig 20. Heat-map analysis of the PAV for pressure.**

model has a 0.2791% success rate. Ultimately, with an average efficiency rate of 0.9167 percent, the superior artificial neural network classification tree. The mean F1 value of the decision mehod is 0.949%. Whenever the neighbouring categorization that is similar reaches a mean of 0.9402%, the naive Bayes model F1 score rises by 0.9487%, while the decision method F1 value lowers by 0.004% explored to the optimized model. The ensemble categorization has an F1 value of 0.8178. Ultimately, the superior neural model tree has a mean accuracy rate of 0.9531 percent. The decision tree has an accuracy of 0.9837% on mean. Whenever the closest neighbour classifier tree has a 0.9593% mean accuracy, the decision model has a 0.08% worse precision than the optimizable categorization. As mentioned before, the ensemble model has a

**Table 6. Confusion Matrix Analysis of a ventilator system of pressure.**

| Trees | CPAP | | | | PAV | | | |
|---|---|---|---|---|---|---|---|---|
| | $t_p$ | $t_n$ | $f_p$ | $f_n$ | $t_p$ | $t_n$ | $f_p$ | $f_n$ |
| DT | 106 | 11 | 17 | 11 | 118 | 11 | 5 | 11 |
| OBT | 123 | 0 | 0 | 22 | 121 | 11 | 2 | 11 |
| NBT | 103 | 11 | 20 | 11 | 118 | 12 | 5 | 10 |
| NeNT | 64 | 16 | 59 | 6 | 109 | 12 | 14 | 10 |
| ET | 104 | 11 | 19 | 11 | 79 | 12 | 44 | 10 |
| NNT | 120 | 11 | 3 | 11 | 121 | 12 | 2 | 10 |

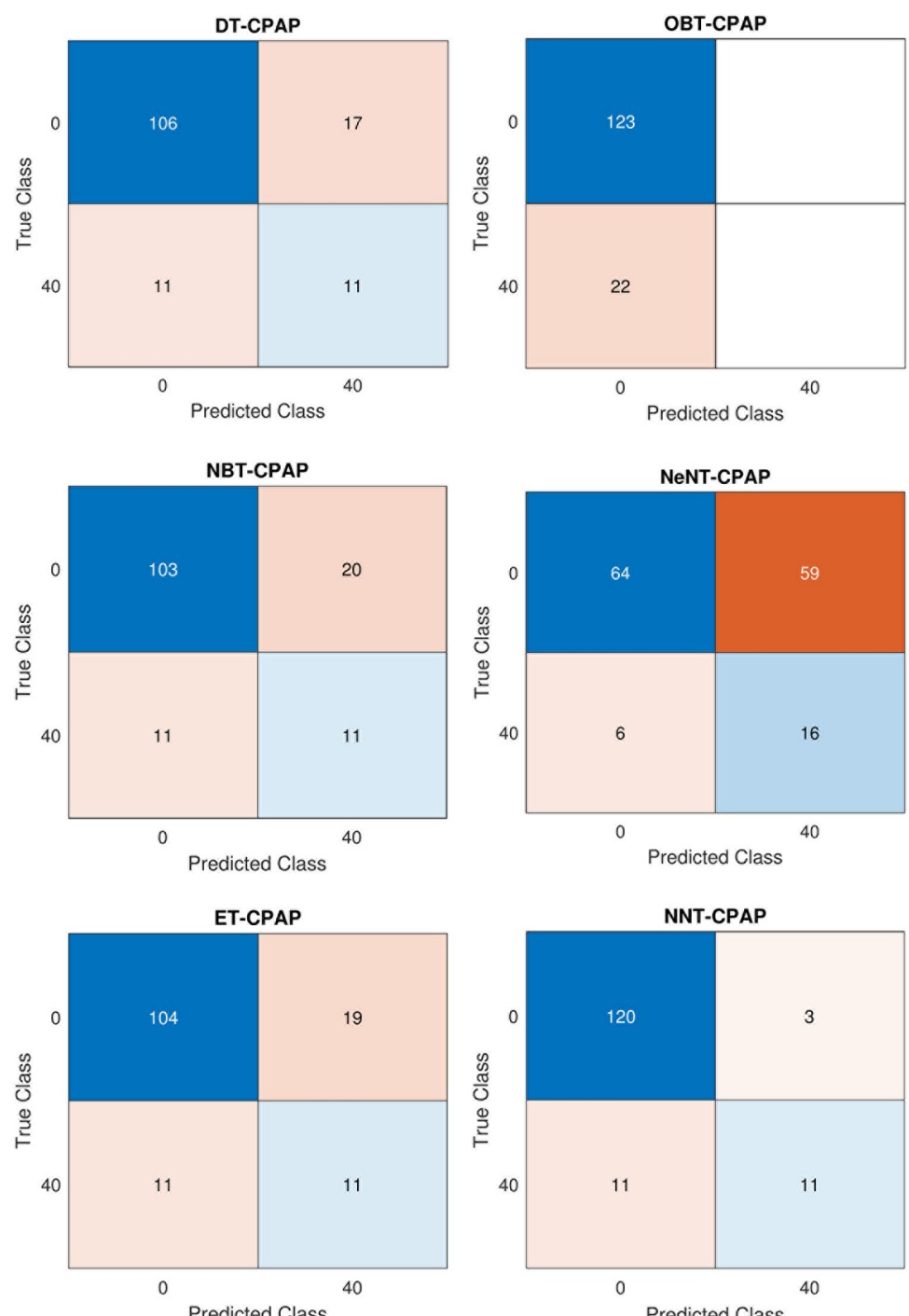

**Fig 21. Confusion Matrix in CPAP Mode for Pressure (a) DT (b) OBT (c) NBT (d) NeNT (e) ET (f) NNT.**

higher specificity and accuracy than the another tree. Finally, the better optimizable model has a 0.9919% preciseness.

Compared to the optimizable tree, the closest neighbour categorization achieves an mean of 0.821%, the naïve model F1 value rises by 0.632%, despite the decision method F1 value

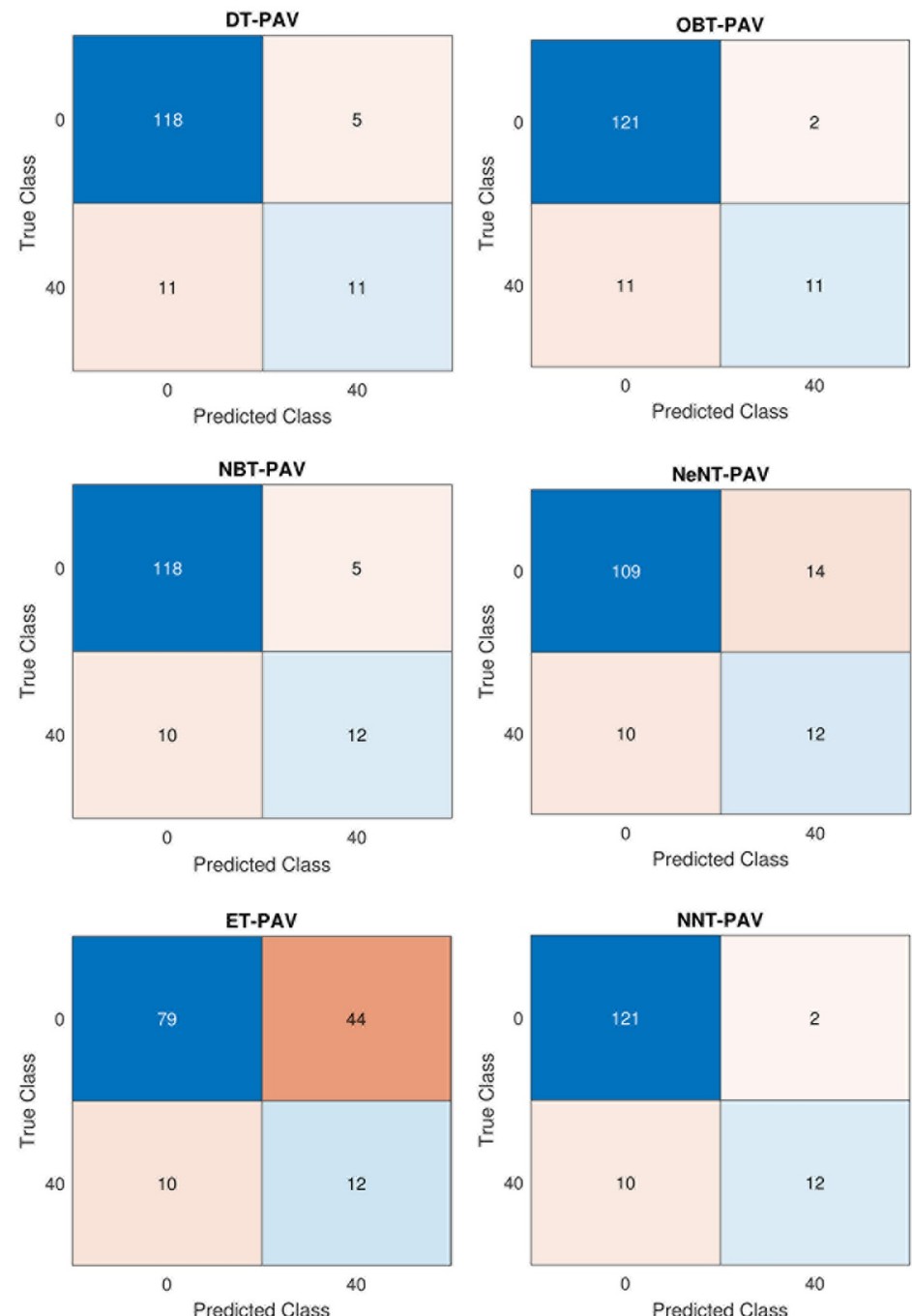

**Fig 22. Confusion Matrix in PAV Mode for pressure (a) DT (b) OBT (c) NBT (d) NeNT (e) ET (f) NNT.**

decreases by 0.201%. In conclusion, the better neural model has a mean accuracy of 0.689%. While the closest neighbour classifier tree has a mean precision of 0.751%, the naïve model has a higher precision; nevertheless, the decision tree has a worse accuracy than the optimizable tree, reducing by 0.1915%. The mean of the ensembled categorization is 0.629. Ultimately, the

better neural model had a mean of of 0.854%. The naive Bayes and nearest neighbour classifiers obtain a mean accuracy of 89.7%. The ensemble classifier tree is 71.7 percent accurate. Finally, with an average accuracy rate of 91.7%, the suggested neural network tree is the better-performing classifier. The neural network tree takes 4.0832 seconds to train, which is approximately longer than the 4.739 seconds necessary to create an ensemble tree or the 1.269 seconds required to complete the closest neighbour tree as the nearest model gains an mean of 0.9225%, the decision method accuracy rises by 0.2007% as explored to the optimized model. While the specificity of the naive model decreases by 0.7059% when a model of 0.7059% is produced employing the closest neighbour model, specificity increases by 0.9167%.

The ensemble method has a success rate of 0.2791%. The closest neighbour categorization shows a mean of 0.9402%. Adding the nearest neighbour method increases the F1 value of the naive Bayes modelby 0.9487%. However, the decision method F1 value drops by 0.004% contrasted to the optimized tree. The ensemble classifier tree algorithm has an F1 value of 0.8178. The top neural network categorization has an mean accuracy rate of 0.9531%. While the decision model has a mean accuracy of 0.9593%, it has 0.08% less precision than the optimizable tree. Furthermore, the ensemble classifier model has greater specificity and accuracy than the other trees. Finally, the better optimized network categorization has a mean preciseness of 0.9919%.

The ROC curve for the breathing apparatus when utilized for pressure in CPAP mode is depicted in Fig 23. In this automated ventilator system, NBT and OBT show lower precision values on the ROC curve than NeNT. Finally, an accurate and adequate value for NNT has been assigned for the curve. The neural network classifier beat all other methods on the ROC curve.

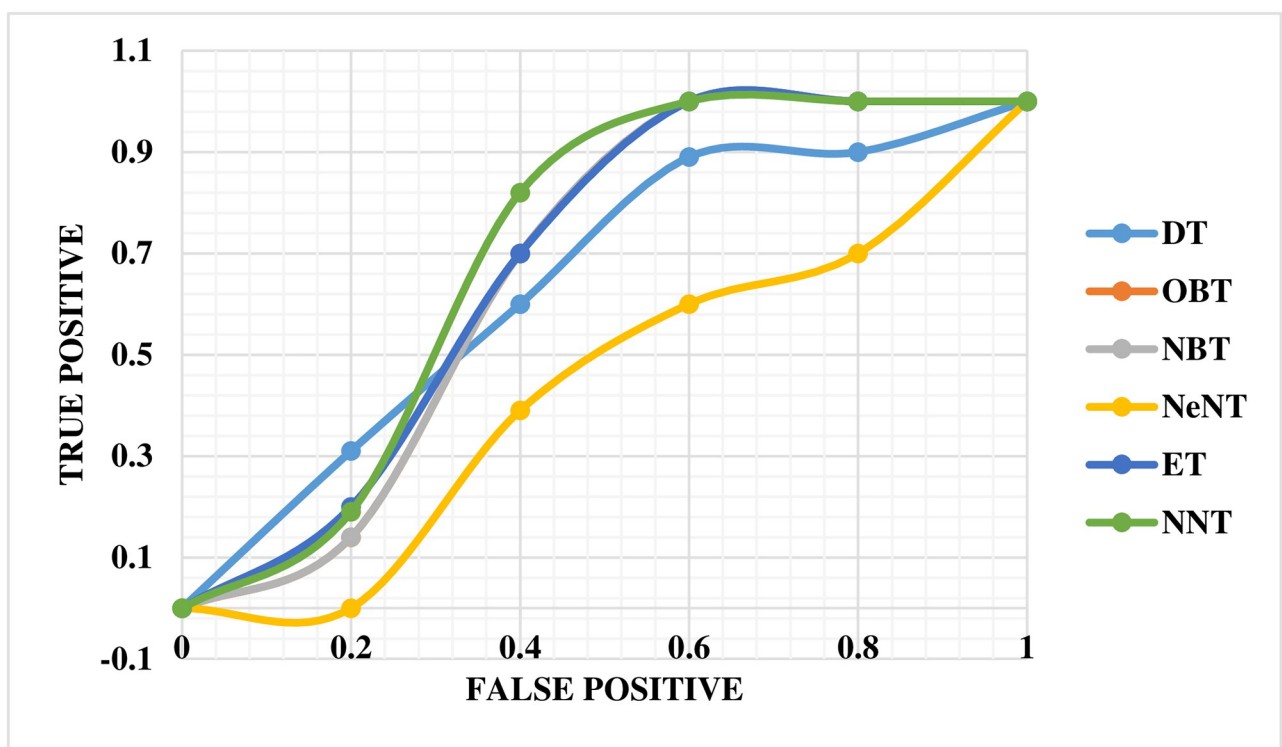

**Fig 23. ROC curve for pressure in CPAP mode.**

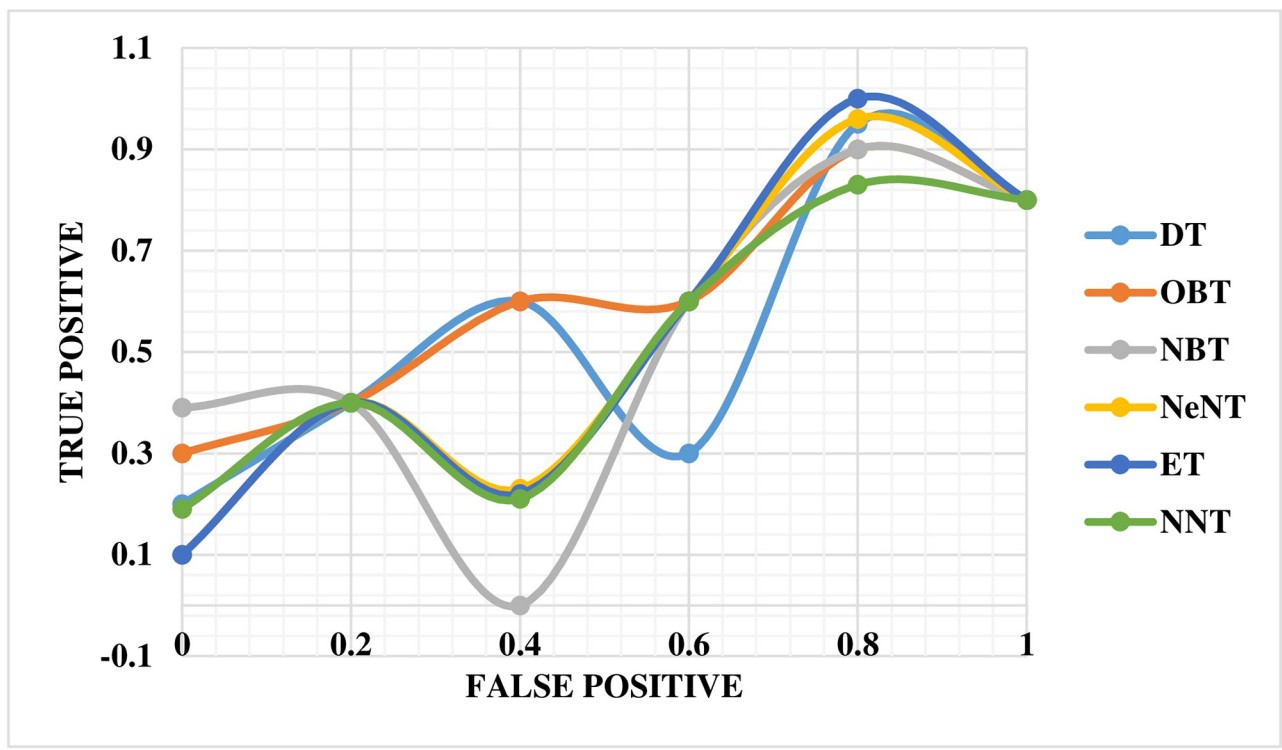

**Fig 24. ROC curve for pressure in PAV mode.**

Fig 24 depicts the ROC curve for the breathing apparatus employed in volume mode in PAV. It is worth noting that the exact worth on the ROC curve for NeNT in this automated ventilation system is lower than that of OBT. The curve has also been allocated the exact and suitable value for NNT. It is critical to emphasize that the neural network classifier outperformed all other techniques in terms of the ROC curve.

The accuracy of both CPAP and PAV modes was fulfilled using the classifier confusion matrix concept, as given in the performance metrics equations for pressure and volume control in the ventilator system.

## 6 Summary and conclusions

The suggested research is extensively summarized in the first part of this section, then the summary outlining the potential applications of the current study.

### 6.1 Summary

The ventilator volume and pressure are controlled via a new cyclic feedback ILC-PID. Furthermore, additional evaluation and verification are performed utilizing a range of AI methods to assess the usefulness of the suggested paradigm, which determines the outlined model of breathing apparatus. The outcome of the ILC-PID method will be provided to the classifier methods in MATLAB Simulation. for different pressure and volume criteria of 20 cm of $H_2O$. The machine learning models' performance was evaluated using accuracy by confusion matrix, training duration, sensitivity, specificity, and F1 value. Tables 3 and 5 of volume and pressure of 20 cm $H_2O$ illustrate the results.

The following points highlight some of the article's significant contributions:

1. A unique ILC-PID method is being researched for more efficient respiratory pressure and volume.

2. For 20 cm $H_2O$ of pressure, the highest accuracy categorization tree for neural networks had a mean preciseness of 88.2% in CPAP mode and 91.7% in PAV mode by using confusion matrix.

3. For 20 cm $H_2O$ of volume, the neural model had a mean accuracy rate of 81.6% in CPAP mode and 84.59% in PAV mode.

4. In most comparisons, the outlined neural network classifier outperforms others in Performance metrics.

5. The suggested neural network has the maximum ROC curve performance in both CPAP and PAV modes.

## 6.2 Conclusion

The time-dependent dissemination of lung pressure and volume in response to respirator force is influenced by modifications in respiratory adherence, capability, and pulmonary recruitment. A unique ILC-PID regulates the pressure and volume in the mechanical ventilator throughout the required period. The outlined method reaches the needed elevated pressure and volume, which aids in maintaining the respiratory structure to produce the appropriate pressure for breathing among humans. Furthermore, multiple machine learning based classifiers such as DT, OBT, NBT, NeNT, ET, and NNT trees for varied volume and pressure of 20 cm $H_2O$ are utilized to forecast and select the optimal model for specific ventilator modes such as CPAP and PAV. Compared to the current techniques, the outlined narrow model has a mean of 88.2% in CPAP mode, 91.7% in PAV mode for pressure, 81.6% in CPAP mode and 84.59% in PAV mode for volume by using confusion matrix. The system learned excellent and consistent achievement owing to the merging of modelled record with health record throughout the instruction of neural models. Subsequent investigations will focus on improving and getting external validation for the method used in new clinical decision support instruments.. This will allow us to understand the clinical consequences better and aid clinicians in establishing stronger and tailored treatment with airflow programs. Furthermore, the suggested ILC-PID controller's stand-alone module will be created and coupled with the ventilators for improved performance.

## Supporting information

**S1 Data.**
(XLS)

## Author Contributions

**Conceptualization:** Anitha T.

**Data curation:** Anitha T.

**Formal analysis:** Anitha T., Gopu G., Arun Mozhi Devan P.

**Investigation:** Anitha T., Gopu G., Arun Mozhi Devan P.

**Methodology:** Anitha T., Arun Mozhi Devan P., Maher Assaad.

**Project administration:** Gopu G., Maher Assaad.

**Resources:** Gopu G.

**Software:** Anitha T.

**Supervision:** Gopu G., Maher Assaad.

**Validation:** Gopu G.

**Visualization:** Anitha T., Arun Mozhi Devan P., Maher Assaad.

**Writing – original draft:** Anitha T., Arun Mozhi Devan P.

**Writing – review & editing:** Anitha T., Arun Mozhi Devan P.

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
