## [Decision Letter · Decision Letter 0]

18 Oct 2023

PONE-D-23-25069Classifier Machine Learning Algorithm for Ventilator Mode Selection, Pressure and Volume ControlPLOS ONE

Dear Dr. Thilakar,

Thank you for submitting your manuscript to PLOS ONE. After careful consideration, we feel that it has merit but does not fully meet PLOS ONE’s publication criteria as it currently stands. Therefore, we invite you to submit a revised version of the manuscript that addresses the points raised during the review process.

We look forward to receiving your revised manuscript.

Kind regards,

Praveen Kumar Donta, Ph.D.

Academic Editor

PLOS ONE

Journal Requirements:

Reviewers' comments:

Reviewer's Responses to Questions

**Comments to the Author**

1. Is the manuscript technically sound, and do the data support the conclusions?

Reviewer #1: Yes

Reviewer #2: Yes

Reviewer #3: Yes

2. Has the statistical analysis been performed appropriately and rigorously? 

Reviewer #1: Yes

Reviewer #2: N/A

Reviewer #3: Yes

3. Have the authors made all data underlying the findings in their manuscript fully available?

Reviewer #1: Yes

Reviewer #2: Yes

Reviewer #3: Yes

4. Is the manuscript presented in an intelligible fashion and written in standard English?

Reviewer #1: Yes

Reviewer #2: Yes

Reviewer #3: Yes

5. Review Comments to the Author

Reviewer #1: 1. Author need to synchronize the usage of the classifier terms in this research similar to the term written in Fig 3.

2. Author need to further elaborate the steps/activities involves in the classifier framework (fig 4) and relate with the discussion in line 205.

3. Please modify the section number in the summarization of introduction (end paragraph of introduction section) accordingly.

4. Please write/refer to the exact figure no and table no and not general statement of figure and number (e.g line 341)

Reviewer #2: 1. Hyperparameters need to be mentioned for ML classifiers.

2. Whether 5 fold or 10 fold validation performed?

3.Image quality need to be improved.

4. Contributions should have crisp points.'

5. Fig 5 is general...hence can be removed.

6. Discussion of proposed work results with existing literature works needs to be improved.

Reviewer #3: The article is well written in terms of problem specifications, modeling, and results according to the different machine learning algorithms. The article is also suitable in terms of scientific contributions.

Only thing which I have noticed to be rectified in Figures, 12, 13, 17 and 18 for term 'Decision Trees'. Which has to be corrected. Else is fine.

6. PLOS authors have the option to publish the peer review history of their article (what does this mean?). If published, this will include your full peer review and any attached files.

Reviewer #1: No

Reviewer #2: No

Reviewer #3: No

---

## [Author Response · Author response to Decision Letter 0]

24 Nov 2023

Reviewer 1:

1. Author need to synchronize the usage of the classifier terms in this research similar to the term written in Fig 3. 

Author Response: We are grateful for your recognition of our research article. Your review comments are highly esteemed as they allow us to enhance the quality of our work. In the revised article, as per the suggestions, terms are synchronized and addressed everywhere. Further, the terms are abbreviated and given in the abbreviation section. (See Page 10, highlighted part, Lines 345 to 348.)

2. Author need to further elaborate the steps/activities involves in the classifier framework (fig 4) and relate with the discussion in line 205. 

Author Response: Thank you for your valuable suggestions. In the updated article, we have added the activities involved in every phase of the proposed framework with the respective correlation of each step.

(See Page 7, highlighted sentences, Lines 210 to 231.)

3. Please modify the section number in the summarization of introduction (end paragraph of introduction section) accordingly. 

Author Response: We greatly appreciate your suggestions. We have updated the Introduction section number. (See Page 1, Section title.)

4. Please write/refer to the exact figure no and table no. and not general statement of figure and number (e.g line 341)

Author Response: As per the suggestion, we have updated the table and figure calling in the revised article. (See Pages 10 and 11, Lines 375-378.)

Reviewer 2:

1. Hyperparameters need to be mentioned for ML classifiers. 

Author Response: Thanks for your kind suggestions, and your review comments are highly esteemed as they allow us to enhance the quality of our work. In the updated article, the hyperparameters used for the ML classifiers are updated and given in the results and discussion section. (See Page 10, highlighted sentences, Lines 339 to 345.)

2. Whether 5 fold or 10 fold validation performed?

Author Response: In our research study, we have used 5 folds for performance validation.

3. Image quality need to be improved 

Author Response: Thanks for your kind suggestions. We have generated all our results in EPS format. However, the submission portal only allows PDF or TIFF, so the conversion caused this issue. However, we use PDF format in the final upload to avoid this issue.

4.Contributions should have crisp points.

Author Response: Thanks for your kind suggestions. We have updated the article's contributions and significance in this updated version. (See Page 4, highlighted sentences, Lines 104 to 123.)

5. Fig 5 is general...hence can be removed. 

Author Response: As per the suggestions, we have removed Fig 5.

6. Discussion of proposed work results with existing literature works needs to be improved. 

Author Response: The proposed ILCPID controller technique is compared with the existing PID controller. Also, the classifiers are compared with different methods like decision trees, optimizable, naive Bayes, nearest neighbour, ensemble classifier, and neural network. In addition, we have further improved the existing literature works.

Reviewer 3:

1. The article is well written in terms of problem specifications, modeling, and results according to the different machine learning algorithms. The article is also suitable in terms of scientific contributions. 

Author Response: Thank you for acknowledging our research article. We greatly appreciate your review comments and believe they will allow us to improve the quality of our work. Your feedback is invaluable in helping us to refine and enhance our research.

2. Only thing which I have noticed to be rectified in Figures, 12, 13, 17 and 18 for term 'Decision Trees'. Which has to be corrected. Else is fine. 

Author Response: Thanks for your kind suggestions. In Figures 12, 13, 17 and 18, in order to avoid the repetition of the word “trees” in all the ML classifiers, we have kept them on the vertical axis for better readability. However, we have addressed this and rewritten it in the results and discussion section. (See Page 10, highlighted part, Lines 345 to 348.)

---

## [Decision Letter · Decision Letter 1]

8 Dec 2023

PONE-D-23-25069R1Classifier Machine Learning Algorithm for Ventilator Mode Selection, Pressure and Volume ControlPLOS ONE

Dear Dr. Thilakar,

Thank you for submitting your manuscript to PLOS ONE. After careful consideration, we feel that it has merit but does not fully meet PLOS ONE’s publication criteria as it currently stands. Therefore, we invite you to submit a revised version of the manuscript that addresses the points raised during the review process.

We look forward to receiving your revised manuscript.

Kind regards,

Praveen Kumar Donta, Ph.D.

Academic Editor

PLOS ONE

**Additional Editor Comments:**

Please consider reviewers comments carefully and resubmit the paper with em or ugh improvements.

Reviewers' comments:

Reviewer's Responses to Questions

**Comments to the Author**

1. If the authors have adequately addressed your comments raised in a previous round of review and you feel that this manuscript is now acceptable for publication, you may indicate that here to bypass the “Comments to the Author” section, enter your conflict of interest statement in the “Confidential to Editor” section, and submit your "Accept" recommendation.

Reviewer #2: All comments have been addressed

Reviewer #3: All comments have been addressed

Reviewer #4: (No Response)

2. Is the manuscript technically sound, and do the data support the conclusions?

Reviewer #2: Yes

Reviewer #3: Yes

Reviewer #4: No

3. Has the statistical analysis been performed appropriately and rigorously? 

Reviewer #2: N/A

Reviewer #3: Yes

Reviewer #4: No

4. Have the authors made all data underlying the findings in their manuscript fully available?

Reviewer #2: Yes

Reviewer #3: Yes

Reviewer #4: Yes

5. Is the manuscript presented in an intelligible fashion and written in standard English?

Reviewer #2: Yes

Reviewer #3: Yes

Reviewer #4: No

6. Review Comments to the Author

Reviewer #2: The authors have addressed all the comments given by the reviewers . Hence the article can be accepted in this current form.

Reviewer #3: In the article, "Classifier Machine Learning Algorithm for Ventilator Mode Selection, Pressure and

Volume Control" authors have addressed all the review queries. Now, I recommend the article for acceptance.

Reviewer #4: The paper proposes the existing ML techniques for the Ventilator Mode Selection. A simulation was developed to validate the proposed framework. However, there are many issues about this work. It is hard to follow, and the contribution is unclear.

In its current form, the paper only shows a more or less trivial approach. A better description of the novelty of this work is required right from the introduction. Furthermore, considering the current advances on ML, it becomes hard to understand the interest in this topic. In my opinion, the authors should take some efforts for convincing the reader of the interest and validity of their approach.

Authors need to deeply review both grammar and the organization of the paper. Generally speaking, the paper has many grammatical issues with several sentences hard to comprehend. Thus, it cannot be accepted without profound proof-reading. I recommend a whole paper revision by a native English speaker.

Other Comments:

In the title -> "Classifier Machine Learning Algorithm": "Classifier" and "Machine Learning Algorithm" are both terminologies used interchangeably in the research. Both are nouns, and therefore, the title is vague and does not clearly represent the content.

The abstract must present results for other classifiers that have been tested alongside NN.

The word "best" is used to indicate better performance in the article. I believe nothing is the best; however, when comparing techniques, we can talk about "better" performance of one technique over another.

The contribution listed in Line# 104 states a significant contribution. This is a very strong statement. I suggest the authors clearly articulate the contribution and provide statistical proof for general scenarios when using the term "significant."

Line 125: How can authors say "most frequently"? Is there any supporting data? Furthermore, in the next line, authors have selected CPAP and PAV modes but give no justification for choosing these among a number of existing modes.

Line 515: Using "significantly" implies that the facts are generally proven using statistical techniques. Otherwise, very strong justification would be required to claim that it gives the same results in all cases.

The authors need to familiarize themselves with terms like Classifier, Classifier Algorithm, Machine Learning Algorithm, etc. and then check the write-up against the terminologies used.

Section 3.3: I could not get a clear picture of this section. The explanation, the storyline, the organization, and the relevance with Fig 4. There is no clear flow of explanation given.

Table 3 and Table 4: We need to understand why one technique performs better and why the other performs poorly. It is not just about translating facts and figures into text.

Overall, it seems that the concept of ML techniques is weakly presented in the article. Starting from the ML background study, then the selection of techniques, implementation (experimental setup), and ending with experimental analysis—all these phases lack strong storylines and knowledge.

7. PLOS authors have the option to publish the peer review history of their article (what does this mean?). If published, this will include your full peer review and any attached files.

Reviewer #2: No

Reviewer #3: **Yes: **Dr. Sanjiv Kumar Jain

Reviewer #4: No

---

## [Author Response · Author response to Decision Letter 1]

7 Feb 2024

All the Reviewers comments (Round 2) are addressed and revised in the updated article. Further, I have attached the manuscript with highlighted changes PDF, updated latex file (.tex) file and removed all the old files (Kept the official letters and dataset in the files).

---

## [Editor Report · Decision Letter 2]

13 Feb 2024

Machine Learning Algorithm for Ventilator Mode Selection, Pressure and Volume Control

PONE-D-23-25069R2

Dear Dr. Thilakar,

We’re pleased to inform you that your manuscript has been judged scientifically suitable for publication and will be formally accepted for publication once it meets all outstanding technical requirements.

Kind regards,

Praveen Kumar Donta, Ph.D.

Academic Editor

PLOS ONE

Additional Editor Comments (optional):

No further comments.
---

## [Editor Report · Acceptance letter]

1 Mar 2024

PONE-D-23-25069R2 

PLOS ONE

Dear Dr. Thilakar, 

I'm pleased to inform you that your manuscript has been deemed suitable for publication in PLOS ONE. Congratulations! Your manuscript is now being handed over to our production team.

Kind regards, 

on behalf of

Dr. Praveen Kumar Donta 

Academic Editor

PLOS ONE